# HiNeRV: Video Compression with Hierarchical Encoding-based Neural Representation

**Ho Man Kwan**†, **Ge Gao**†, **Fan Zhang**†, **Andrew Gower**‡, **David Bull**†
† Visual Information Lab, University of Bristol, UK
‡ Immersive Content & Comms Research, BT, UK
{hm.kwan, ge1.gao, fan.zhang, dave.bull}@bristol.ac.uk,
andrew.p.gower@bt.com

## Abstract

Learning-based video compression is currently a popular research topic, offering the potential to compete with conventional standard video codecs. In this context, Implicit Neural Representations (INRs) have previously been used to represent and compress image and video content, demonstrating relatively high decoding speed compared to other methods. However, existing INR-based methods have failed to deliver rate quality performance comparable with the state of the art in video compression. This is mainly due to the simplicity of the employed network architectures, which limit their representation capability. In this paper, we propose HiNeRV, an INR that combines light weight layers with novel hierarchical positional encodings. We employs depth-wise convolutional, MLP and interpolation layers to build the deep and wide network architecture with high capacity. HiNeRV is also a unified representation encoding videos in both frames and patches at the same time, which offers higher performance and flexibility than existing methods. We further build a video codec based on HiNeRV and a refined pipeline for training, pruning and quantization that can better preserve HiNeRV's performance during lossy model compression. The proposed method has been evaluated on both UVG and MCL-JCV datasets for video compression, demonstrating significant improvement over all existing INRs baselines and competitive performance when compared to learning-based codecs (72.3% overall bit rate saving over HNeRV and 43.4% over DCVC on the UVG dataset, measured in PSNR). [1]

## 1   Introduction

Implicit neural representations (INRs) have become popular due to their ability to represent and encode various scenes [37], images [45] and videos [45, 11]. INRs typically learn a coordinate to value mapping (e.g. mapping a pixel or voxel index to its color and/or occupancy) to support implicit reconstruction of the original signal. While these representations are usually instantiated as multilayer perceptrons (MLPs), existing MLP-based network can only represent video content with a low reconstruction quality and speed [11]. To address this limitation, recent works have employed Convolutional Neural Networks (CNNs) to perform a frame index to video frame mapping [11, 29, 5, 25, 10]. These CNN-based INRs are capable of reconstructing video content with higher quality and with a faster decoding speed, when compared to MLP-based approaches [45]. When using INRs for encoding videos, video compression can then be achieved by performing model compression for the individual input video. However, existing INR-based algorithms remain significantly inferior to state-of-the-art standard-based [52, 47, 8] and learning-based codecs [26, 41, 27, 28, 35]. For example, none of these INR-based codecs can compete with HEVC x265 [2] (*veryslow* preset).

---

[1]Project page: `https://hmkx.github.io/hinerv/`

37th Conference on Neural Information Processing Systems (NeurIPS 2023).

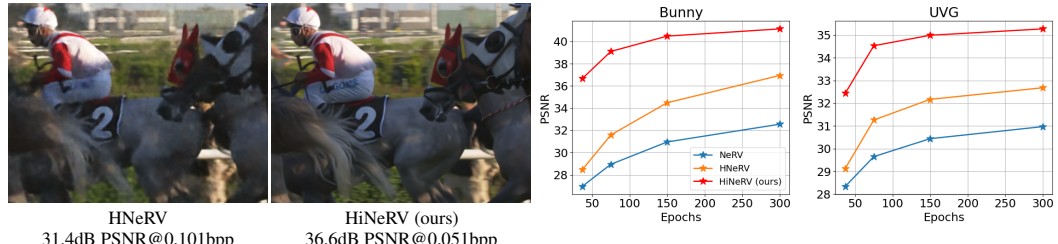

Figure 1: (Left) Visual comparison between HNeRV [10] and HiNeRV (ours) for compressed content (cropped). HiNeRV offers improved visual quality with approximately half the bit rate compared to HNeRV (PSNR and bitrate values are for the whole sequence). (Right) Video regression with different epochs for a representation task. HiNeRV (ours) with only 37 epochs achieves similar reconstruction quality to HNeRV [10] with 300 epochs.

Most INR-based models for videos [11, 29, 5, 25, 10] employ conventional convolutional layers or sub-pixel convolutional layers [42], which are less parameter efficient, and hence limit representation capacity within a given storage budget. In addition, most existing work employs Fourier-based positional encoding [37]; this has a long training time and can only achieve sub-optimal reconstruction quality [11, 29, 5]. In video compression, the training of INR models is equivalent to the encoding process, implying that most INR-based codecs require a long encoding runtime to obtain a satisfactory rate-quality performance [11]. However, some recent non-video INR models have utilized feature grids or a combination of grids and MLPs as the representation to speed up the convergence of INRs; this has improved the encoding speed by several orders of magnitude [15, 48, 38, 9].

In this paper, we propose a new INR model based on Hierarchically-encoded Neural Representation for video compression, HiNeRV. We replace commonly used sub-pixel conventional layers [42] in existing INRs for upsampling [11, 29, 5, 25, 10] by a new upsampling layer which embodies bilinear interpolation with hierarchical encoding that is sampled from multi-resolution local feature grids. These local grids offer increased parameter efficiency, as the number of parameters increases with the upsampling factor instead of the resolution. Moreover, the network is primarily based on MLPs and depth-wise convolutional layers (rather than dense convolutional layers). This enhances the representation capability and maximizes the performance for a given parameter count. This architectural design allows us to build a much deeper and wider network which offers a significantly better video encoding performance when compared to state-of-the-art INR-based coding approaches.

Furthermore, instead of learning a frame- [11] or patch-wise [5] representation, we show that by simply training with overlapped patches, HiNeRV can be seamlessly switched between both representation types, achieving a unified representation with improved performance compared to both frame-based and patch-based settings. This also provides flexibility for hardware implementation, where encoding and decoding processes can be performed either using frames to optimize the computational complexity, or as patches to minimize the memory footprint.

To achieve competitive coding performance, we also refine the model compression pipeline in [11], where pruning and fine-tuning are followed by model training, before quantization is applied. First, we used an adaptive pruning technique to reduce the negative impact of model pruning. Secondly, quantization-aware training is applied for fine-tuning the model performance before quantization. This enables lower bit depth quantization which achieves an improve rate-distortion trade-off.

The proposed method has been tested against existing INR-based video coding methods and state-of-the-art conventional and learning-based video codecs on the UVG [36] and MCL-JCV [51] datasets. Notwithstanding the fact that HiNeRV has not been end-to-end optimized for video compression (i.e. pruning, quantization and entropy coding are not included in the training loop), it still significantly outperforms all INR-based methods (e.g., 72.3% overall BD-rate gain over HNeRV on the UVG database, measured in PSNR), and is also competitive compared with existing conventional and end-to-end learning-based video codec (e.g., 43.4% over DCVC, 38.7% over x265 *veryslow*).

The primary contributions of this work are summarized below:

1) We propose HiNeRV, a new INR employing hierarchical encoding based neural representation.

2) We employ a unified representation by adding padding, which trades a small computation overhead for additional flexibility and performance gain.

3) We build a video codec based on HiNeRV and refine the model compression pipeline to better preserve the reconstruction quality of INRs by using adaptive pruning and quantization-aware training.

4) The compression performance of the proposed method is superior to existing INR models, and is comparable to many conventional/learning-based video coding algorithms. As far as we are aware, it is the first INR-based codec to significantly outperform HEVC (x265 *veryslow*) [39].

## 2   Related work

### 2.1   Video compression

Video compression has long been a fundamental task in the field of computer vision and multimedia processing. As alternatives to the already popularized conventional codecs such as H.264/AVC [52], H.265/HEVC [47], and H.266/VVC [8], there has been a rapid increase in the adoption of deep learning techniques for video compression in recent years. This has typically involved replacing certain modules (e.g., motion compensation [3, 54], transform coding [55, 16] and entropy coding [6]) in the conventional pipeline with powerful learning-based models [53, 35].

In contrast, there has also been significant activity focused on the development of new coding architectures which allow end-to-end optimization. Lu et al. [32] proposed DVC that was further extended to enable major operations in both the pixel and the feature spaces [21, 20]. An alternative approach has focused on conditional [26, 30] instead of predictive coding to reduce the overall bitrate by estimating the probability model over several video frames. Furthermore, the characteristics of the differentiable frameworks have been exploited by [19, 49, 44], where both encoder and decoder (typically signaled by a model stream containing updated parameters) are overfitted to the video data during the evaluation to further enhance compression efficiency.

While effective, with some recent work [53, 43, 28] claiming to outperform the latest compression standards, these methods still follow the pipeline of conventional codecs, which may constrain the development of neural video compression methods. Moreover, learning-based video compression methods tend to be much more computational complex and often yield much slower decoding speed than conventional codecs. This often renders them impractical for real-time applications, especially considering the prevalence of high-quality and high-resolution videos consumed nowadays.

### 2.2   Implicit neural representation

Implicit neural representations (INRs) are being increasingly used to represent complicated natural signals such as images [45, 12, 14, 23], videos [45, 11], and vector-valued, volumetric content [37]. This type of approach benefits from incorporating positional encoding - a technique that embeds the positional input into a higher-dimensional feature space. Periodic functions [45, 33, 37] have first been utilized to improve the network's capability for learning high frequency information, and grid features [48, 38, 9] have then been applied to address their slow convergence speed and further improve the reconstruction quality.

More recently, Neural Representations for Videos (NeRV) [11] has re-formulated the INR for video signals to be frame-wise, achieving competitive reconstruction performance with very high decoding speed. NeRV approaches have inspired a trend of utilizing CNNs to encode the RGB values of videos with 1D frame coordinates [11, 29, 25, 10] or with 3D patch coordinates [5], and have demonstrated promise in various video tasks, including denoising [11], frame interpolation [11, 29, 25, 10], inpainting [5, 10], super-resolution [13] and video compression [11, 29, 5, 25, 10].

When INRs are applied for image and video compression, they typically convert the signal compression task into a model compression problem by incorporating weight pruning, quantization and entropy coding [17]. NeRV [11] and related works [29, 5, 25, 10] adopt the above approach for video compression. Although these have demonstrated fast decoding capability, they do not yet achieve a rate-distortion performance comparable to either conventional or learning-based codecs.

## 3   Method

Following the approach adopted in previous work [45, 11], we consider a video regression task where a neural network encodes a video $V$ by mapping coordinates to either individual frames, patches or

pixels, where $V \in \mathbb{R}^{T \times H \times W \times C}$, $T$, $H$, $W$ and $C$ are the number of frames in $V$, the height, the width and the number of channels of the video frames, respectively.

Fig. 2 (top) illustrates the high level structure of the proposed model, HiNeRV, which contains a base encoding layer, a stem layer, $N$ HiNeRV blocks, and a head layer. In HiNeRV, each RGB video frame is spatially segmented into patches of size $M \times M$, where each patch is reconstructed by one forward pass. The model first takes a patch coordinate $(i, j, t)$ to compute the base feature maps $X_0$ with size $M_0 \times M_0 \times C_0$. Here we always refer the coordinates to the integer index, such that $0 \leq t < T$, $0 \leq j < \frac{H}{M}$ and $0 \leq i < \frac{W}{M}$. The following $N$ HiNeRV blocks then upsample and process the feature maps progressively, where the $n$-th block produces the intermediate feature maps $X_n$ that have the size $M_n \times M_n \times C_n$ ($M_N = M$). Finally, a head layer is used to project the feature maps to the output, $Y$, with the target size $M \times M \times C$.

## 3.1 Base encoding and stem

HiNeRV first maps the input *patch* coordinates into the base feature maps, $X_0$, by

$$X_0 = F_{stem}(\gamma_{base}(i, j, t)). \tag{1}$$

To compute the base feature maps, we first calculate the *pixel* coordinates (related to the corresponding video frame) of the patch. For a patch with size $M_0 \times M_0$, the frame-based pixel coordinates $(u_{frame}, v_{frame})$ can be computed by the patch-based pixel coordinates $(u_{patch}, v_{patch})$ for $0 \leq u_{patch}, v_{patch} < M_0$, such that $u_{frame} = i \times M_0 + u_{patch}$ and $v_{frame} = j \times M_0 + v_{patch}$. Then, by using the frame-based pixel coordinates, the positional encoding $\gamma_{base}(i, j, t)$ can be interpolated from the learned feature grids [25]. After that, we employ a stem convolutional layer $F_{stem}$ for projecting the feature maps to a desired number of channels $C_0$.

It is noted that most existing INRs for video [11, 29, 5, 10] utilize the Fourier style encoding, i.e. they apply $sin$ and $cos$ functions to map coordinates into the positional encoding. However, such encoding contains only positional information, which requires additional layers (e.g. MLPs) to transform it into informative features. In contrast, we adopt grid-based encoding [25] as they that contain richer information than the Fourier encoding. Specifically, we use the multi-resolution temporal grids that were introduced in FFNeRV [25], where the various feature grids have different temporal resolutions. In FFNeRV, linear interpolation over the temporal dimension is used to obtain a slice that is used as the input feature map. In our case, we utilize both of the frame index and the frame-based coordinates, i.e., $(u_{frame}, v_{frame}, t)$, for interpolating the feature patches.

Although both high temporal resolution and a large number of channels are desirable for enhancing the expressiveness of the feature grids, this can result in greater model sizes and hence higher bitrates when the model is used for compression tasks. To maintain a compact multi-resolution grid, we increase the number of channels when reducing the temporal resolution at each grid level, i.e. the size of a grid is $\lfloor \frac{T_{grid}}{2^l} \rfloor \times H_{grid} \times W_{grid} \times (C_{grid} \times 2^l)$, for $0 \leq l < L_{grid}$. Here $L_{grid}$ is the number of grids and $T_{grid} \times H_{grid} \times W_{grid} \times C_{grid}$ is the size of the first level grid.

## 3.2 HiNeRV block

The obtained base feature maps are then processed by $N$ HiNeRV blocks, which progressively upsample and process the feature maps. Specifically, the $n$-th HiNeRV block, where $0 < n \leq N$, upsamples the input feature maps $X_{n-1}$ with size $M_{n-1} \times M_{n-1} \times C_{n-1}$ through bilinear interpolation $U_n$ with a scaling factor $S_n$, such that $M_n = M_{n-1} \times S_n$. We use bilinear interpolation, mainly due to its low computational cost and its capability to compute smooth upsampled maps. The HiNeRV block then computes a hierarchical encoding $\gamma_n(i, j, t)$, matches its number of channels by a linear layer $F_{enc}$ and adds it to the upsampled feature maps, where $\gamma_n(i, j, t)$ matches the upsampled feature map size (see Section 3.4). Finally, it applies a set of network layers $F_n$ to enhance the representation to obtain the output $X_n$, where we specify the number of layers by $D_n$. For all the HiNeRV blocks except the first one ($1 < n \leq N$), the first layer in $F_n$ also reduces the number of channels by a factor, $R$, such that $C_n = \lfloor \frac{C_0}{R^{n-1}} \rfloor$, to save the computational cost for processing high spatial resolution maps. The output of the $n$-th HiNeRV block, $X_n$, has a size of $M_n \times M_n \times C_n$, and can be written as

$$X_n = F_n(U_n(X_{n-1}) + F_{enc}(\gamma_n(i, j, t))), 0 < n \leq N \tag{2}$$

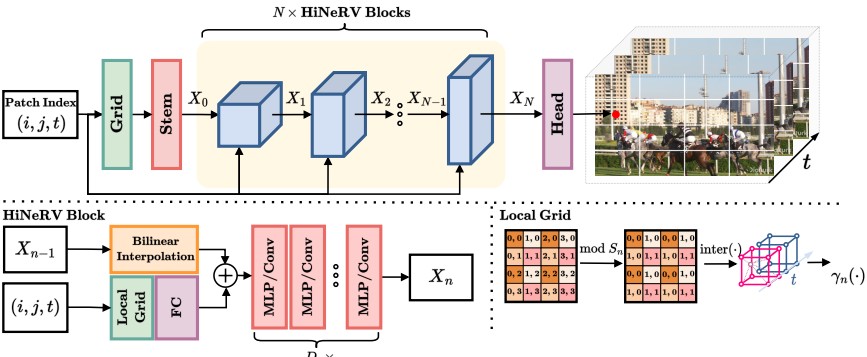

Figure 2: Top: The HiNeRV architecture. Bottom left: The HiNeRV block. HiNeRV block take feature maps $X_{n-1}$ and patch index $(i, j, t)$ as input, upsample the feature maps, enhances it with the hierarchical encoding, then computes the transformed maps $X_n$. Bottom right: The local grid. In HiNeRV, the hierarchical encoding is computed by performing interpolation from the local grid, where the modulo of the coordinates is being used.

Figure 2 (bottom-left) shows the structure of HiNeRV block. Due to its observed superior performance, we employ ConvNeXt [31] as the network block in $F_n$, a combination of the MLP layer with depth-wise convolution. We also apply Layer Normalization [4] before the interpolation and the MLP layers, and we only use shortcut connections when the input and output dimensions are matched.

### 3.3 Head layer

The final output patch $Y$ is computed by applying a linear layer with sigmoid activation, denoted by $F_{head}$, on the output of the $N$-th HiNeRV block,

$$Y = F_{head}(X_N) \tag{3}$$

### 3.4 Upsampling with the hierarchical encoding

Existing NeRV-based approaches [11, 29, 5, 25, 10] use the sub-pixel convolutional layer [42] for feature map up-scaling. However, this has a high parameter complexity: $K^2 \times S^2 \times C_1 \times C_2$, where $K$, $S$, $C_1$ and $C_2$ represent the kernel size, the upsampling factor, the number of input and output channels, respectively. While neighboring features are highly correlated, the convolutional layer does not take this into account and learns a dense weight matrix to perform upsampling. This is inefficient especially when the model size is the concern for tasks like compression [11], because the low parameter efficiency limits the maximum depth and width of the networks, and thus the capacity.

Previous work [11] has shown that bilinear interpolation does not perform as well as convolutional layers. However, we observed that it is actually a better choice when the parameter count is fixed. By performing interpolation, we can utilize the saved parameter budget to build a network with higher capacity. While we can generate high-resolution maps using parameter-free bilinear interpolation, the resulting maps are smoothed, and subsequent neural network layers may struggle to produce high-frequency output from them. One way to model high-frequency signals is to introduce positional encoding [37] during upsampling. As mentioned in Section 3.1, grid-based encoding is preferred due to the good representation performance. However, adopting it to enhance high-resolution feature maps can be costly, as it requires high-dimensional grids. To address this limitation, we introduce a novel grid-based encoding approach called hierarchical encoding, which boosts the upsampling capability of bilinear interpolation without significantly increasing the storage cost.

Unlike normal grid-based encoding, which computes the encoding using global coordinates, hierarchical encoding utilizes local coordinates to encode relative positional information. Specifically, the local coordinate is the relative position of a pixel in the upsampled feature map to its nearest pixel in the original feature map. Because local coordinates have a much smaller range of values, the required feature grid is also much smaller. Moreover, when both the base encoding and hierarchical encoding are used, every position can be represented hierarchically, allowing us to efficiently encode positional information on high-resolution feature maps. During upsampling, the upsampled feature

maps are first produced through bilinear interpolation with an up-scaling factor $S_n$. Then, for all frame-based pixel coordinates $(u_{frame}, v_{frame})$ in the upsampled feature maps, we compute the corresponding local pixel coordinates by $u_{local} = u_{frame} \bmod S_n$ and $v_{local} = v_{frame} \bmod S_n$, and employ them to compute the encoding.

It is noted that the above encoding approach is similar to applying a convolutional layer over a constant feature map. To further enhance the capacity of this encoding, we model the feature grids as multi-temporal resolution grids [25], which can provide richer temporal information, similar to the base encoding. In the $n$-th HiNeRV block, there are $L_{local}$ levels of grids and the $l$-level grid has a size of $\lfloor \frac{T_{local}}{2^l} \rfloor \times S_n \times S_n \times (\lfloor \frac{C_{local}}{R^{n-1}} \rfloor \times 2^l)$. The size of the local grids is scaled with the factor $S_n$ and can be adjusted by the hyper-parameter $T_{local}$ and $C_{local}$. The number of channels of the grids is also scaled in proportion to the width of the HiNeRV block, i.e. by the reduction factor $R$. To obtain the hierarchical encoding, we perform trilinear interpolation by utilizing the frame index $t$ with the local coordinate, i.e., $(u_{local}, v_{local}, t)$, to extract encodings from all levels, then concatenate the encodings and apply a linear layer $F_{enc}$ to match the encoding channels to the feature maps. To distinguish the grids for interpolating the hierarchical encoding from the one for the base encoding, i.e. the *temporal grids*, we refer these grids as the *temporal local grids*, because the hierarchical encoding is interpolated from these grids by using the local pixel coordinates. In Section 4.3, we demonstrated that the hierarchical encoding contributes to the superior performance HiNeRV. The process of upsampling with local encoding is shown in Figure 2 (bottom right).

### 3.5 Unifying frame-wise and patch-wise representations

Recent publications on INR for video can be classified into frame-wise [11, 29, 25, 10] or patch-wise representations [5]. Actually, in many of these networks, the initial feature maps can be easily computed either frame-wise or patch-wise, as the positional encoding depends only on the corresponding pixel coordinates. However, these two types of representations are not switchable because of boundary effects. In HiNeRV, we adopt a simple technique to unify frame-wise and patch-wise representations. When configuring HiNeRV as a patch-wise representation, we perform computation in overlapped patches, where we refer to the overlapped part as paddings, and the amount of padding pixels depends on the network configuration (e.g. the kernel sizes and/or the number of bilinear interpolation/convolutional layers). Such overlapped patches have previously been used for tasks such as super-resolution [22], but have not been applied in NeRV-based methods. When performing encoding in patches without proper padding, missing values for operations such as convolution can result in discontinuities between patch boundaries. Moreover, networks trained in patches do not perform well when inferencing in frames due to boundary effects. In our implementation, we perform intermediate computation with padded patches and crop the non-overlapped parts as the output patches, while with the frame configuration, paddings are not required. By adding paddings, we ensure generation of the same output for both frame-wise and patch-wise configurations.

Although adding paddings introduces additional computational overheads, it does provide the following benefits: (i) it allows parallel computation within the same frame, and reduces the minimum memory requirement for performing forward and backward passes. This shares a design concept with conventional block-based video codecs, and can potentially benefit the compression of immersive video content with higher spatial resolutions (where performing frame-wise calculation may not be supported by the hardware); (ii) It improves the training efficiency when compared to a frame-wise representation, as we can randomly sample patches during training [22], which can better approximate the true gradient and speed up the convergence; (iii) It can also enhance the final reconstruction quality compared to a patch-wise representation without suffering boundary effects. By applying the above technique, HiNeRV can be flexibly configured as either frame-based or patch-based representation without retraining. Our ablation study verifies these benefits by comparing this approach to both the frame-based and patch-based variants (see Section 4.3).

### 3.6 The model compression pipeline

To further enhance the video compression performance, we refined the original model compression pipeline in NeRV [11], which has been used in a series of related work [29, 5, 25, 10]. In [11], model training is followed by weight pruning with fine-tuning to lower model sizes. Post-training quantization is then utilized to reduce the precision of each weight, while entropy coding is further employed for lossless compression. In this work, we made two primary changes to this pipeline:

(i) applying an adaptive weighting to the parameters in different layers for pruning, and (ii) using quantization-aware training with Quant-Noise [46] to minimize the quantization error.

**Model Pruning** in NeRV [11, 29, 5, 25, 10] is typically performed globally with respect to the magnitude of individual weights, with fine-tuning applied after pruning [18]. While various non-magnitude based pruning methods exist (e.g. OBD [24]), here we developed a simple, modified magnitude-based method for network pruning. Intuitively, we assume that wider layers have more redundancy within their parameters - hence pruning these layers tends to have less impact than on the shallower layers. To alleviate the negative impact of pruning, we weight each neuron using both its $\ell 1$ norm and the size of the corresponding layer. Specifically, for a layer with $P$ parameters, $\theta_p, 0 < p \leq P$, we compute a score, $\frac{|\theta_p|}{P^\lambda}$, which reflects the neuron importance, where $\lambda$ is a hyper-parameter (0.5 in our experiments). The pruning is then performed as usual. By applying this weighting scheme, layers with fewer parameters, such as depth-wise convolutional layers and output layers, have less chance to be pruned, while layers in the early stage of the network are more likely to be pruned.

**Weight quantization** plays an important role in model compression, significantly reducing final model size [11]. While the results in [11] have shown that the quantization error is not significant when reducing the weight bit depth to 8 bits, further increasing the quantization level is still meaningful for compression tasks. Unlike other related works [11, 29, 5, 25, 10] that adopted 8 bit quantization, we found that an improved rate-distortion trade-off can be achieved by using 6 bits quantization if a quantization-aware training methodology is applied. In particular, we perform a short fine-tuning with Quant-Noise [46] after weight pruning, which can effectively reduce the quantization error. However, unlike the original implementation, we do not use STE [7] for computing the gradient of the quantized weights due to its inferior performance.

## 4    Experiments

### 4.1    Video representation

To evaluate the effectiveness of the proposed model we benchmarked HiNeRV against five related works: NeRV [11], E-NeRV [29], PS-NeRV [5], FFNeRV [25] and HNeRV [10] on the Bunny [1] ($1280 \times 720$ with 132 frames) and the UVG datasets [36] (7 videos at $1920 \times 1080$ with a total of 3900 frames). For each video, we trained all networks at multiple scales, and kept their number of parameters similar at each scale. Three scales were set up to target the S/M/L scales in NeRV [11] for the UVG datasets, while two different scales XXS/XS together with scale S were configured for the Bunny dataset. We reported the encoding and decoding speeds in frames per second, measured with A100 GPU. The number of parameters corresponding to each scale are reported in Table 1 and 2.

For all models tested, we set the number of training epochs to 300 and batch size (in video frames) to 1. We used the same optimizer as in [11], and employed the same learning objectives for all NeRV-based methods as in the original literature [11, 29, 5, 25, 10]. For HiNeRV, we empirically found that it is marginally better to adopt a larger learning rate of $2e - 3$ with global norm clipping which is commonly used for Transformer-based networks [50]. We used the learning rate of $5e - 4$ which is a common choice for the other networks as in their original literature [11, 29, 5, 25]. We also adopted a combination of $\ell 1$ loss and MS-SSIM loss (with a small window size of $5 \times 5$ rather than $11 \times 11$) for HiNeRV, as we observed that the MS-SSIM loss with a small window size leads to a better performance. For HiNeRV and PS-NeRV [5], we randomly sample patches instead of frames during training, but we scale the number of patches in each batch to keep the same effective batch size. It is noted that the original configuration of HNeRV [10] has an input/output size of $1280 \times 640/1920 \times 960$ with strides (5, 4, 4, 2, 2)/(5, 4, 4, 3, 2), and we pad the frames of HNeRV in order to fit the $1280 \times 720/1920 \times 1080$ videos. This was found to work better than changing the strides in HNeRV. Detailed configuration of HiNeRV is summarized in the *Supplementary Material*.

It can be observed (Table 1 and 2) that our proposed HiNeRV outperforms all benchmarked models in terms of reconstruction quality at each scale on both Bunny [1] and UVG [36] datasets. We also note that HiNeRV performs better than all the other methods on all test sequences (with various spatial and temporal characteristics) in the UVG database, exhibiting better reconstruction quality in terms of PSNR. In particular, for some video sequences where existing INRs performed poorly [10], HiNeRV offers significant improvement (>7.9dB over NeRV on ReadySetGo). While the encoding

Table 1: Video representation results on the Bunny dataset [1] (for XXS/XS/S scales).

| Model | Size | MACs | Encoding FPS | Decoding FPS | PSNR |
|---|---|---|---|---|---|
| NeRV | 0.83M/1.64M/3.20M | 25G/57G/101G | **131.9**/93.6/81.8 | 308.5/229.1/**202.3** | 26.82/29.61/32.56 |
| E-NeRV | 0.88M/1.65M/3.31M | 26G/101G/104G | 104.7/68.0/68.7 | 254.3/175.8/174.8 | 29.03/31.75/36.69 |
| PS-NeRV | 0.90M/1.68M/3.35M | 29G/238G/240G | 90.6/36.1/36.0 | 228.1/96.1/96.0 | 28.47/30.31/34.78 |
| HNeRV | 0.82M/1.66M/3.28M | 23G/48G/94G | 100.0/80.8/64.2 | **317.4/251.6**/192.5 | 31.08/33.68/36.95 |
| FFNeRV | 0.91M/1.66M/3.19M | 26G/58G/102G | 62.1/51.5/47.9 | 108.4/95.2/90.5 | 30.37/33.83/37.01 |
| HiNeRV | 0.77M/1.59M/3.25M | 23G/47G/96G | 37.6/27.7/20.0 | 132.1/103.9/76.7 | **36.37/38.94/41.14** |

Table 2: Video representation results with the UVG dataset [36] (for S/M/L scales). Results are in PSNR. FPS is the encoding/decoding rate.

| Model | Size | MACs | FPS | Beauty | Bosph. | Honey. | Jockey | Ready. | Shake. | Yacht. | Avg. |
|---|---|---|---|---|---|---|---|---|---|---|---|
| NeRV | 3.31M | 227G | **32.4**/90.0 | 32.83 | 32.20 | 38.15 | 30.30 | 23.62 | 33.24 | 26.43 | 30.97 |
| E-NeRV | 3.29M | 230G | 20.7/75.9 | 33.13 | 33.38 | 38.87 | 30.61 | 24.53 | 34.26 | 26.87 | 31.75 |
| PS-NeRV | 3.24M | 538G | 14.7/42.6 | 32.94 | 32.32 | 38.39 | 30.38 | 23.61 | 33.26 | 26.33 | 31.13 |
| HNeRV | 3.26M | 175G | 24.6/**93.4** | 33.56 | 35.03 | 39.28 | 31.58 | 25.45 | 34.89 | 28.98 | 32.68 |
| FFNeRV | 3.40M | 228G | 19.0/49.3 | 33.57 | 35.03 | 38.95 | 31.57 | 25.92 | 34.41 | 28.99 | 32.63 |
| HiNeRV | 3.19M | 181G | 10.1/35.5 | **34.08** | **38.68** | **39.71** | **36.10** | **31.53** | **35.85** | **30.95** | **35.27** |
| NeRV | 6.53M | 228G | **32.0/90.1** | 33.67 | 34.83 | 39.00 | 33.34 | 26.03 | 34.39 | 28.23 | 32.78 |
| E-NeRV | 6.54M | 245G | 20.5/74.6 | 33.97 | 35.83 | 39.75 | 33.56 | 26.94 | 35.57 | 28.79 | 33.49 |
| PS-NeRV | 6.57M | 564G | 14.6/42.0 | 33.77 | 34.84 | 39.02 | 33.34 | 26.09 | 35.01 | 28.43 | 32.93 |
| HNeRV | 6.40M | 349G | 20.1/68.5 | 33.99 | 36.45 | 39.56 | 33.56 | 27.38 | 35.93 | 30.48 | 33.91 |
| FFNeRV | 6.44M | 229G | 18.9/49.3 | 33.98 | 36.63 | 39.58 | 33.58 | 27.39 | 35.91 | 30.51 | 33.94 |
| HiNeRV | 6.49M | 368G | 8.4/29.1 | **34.33** | **40.37** | **39.81** | **37.93** | **34.54** | **37.04** | **32.94** | **36.71** |
| NeRV | 13.01M | 230G | **31.7/89.8** | 34.15 | 36.96 | 39.55 | 35.80 | 28.68 | 35.90 | 30.39 | 34.49 |
| E-NeRV | 13.02M | 285G | 21.0/68.1 | 34.25 | 37.61 | 39.74 | 35.45 | 29.17 | 36.97 | 30.76 | 34.85 |
| PS-NeRV | 13.07M | 608G | 14.1/41.4 | 34.50 | 37.28 | 39.58 | 35.34 | 28.56 | 36.51 | 30.28 | 34.61 |
| HNeRV | 12.87M | 701G | 15.6/52.7 | 34.30 | 37.96 | 39.73 | 35.47 | 29.67 | 37.16 | 32.31 | 35.23 |
| FFNeRV | 12.66M | 232G | 18.4/49.3 | 34.28 | 38.48 | 39.74 | 36.72 | 30.75 | 37.08 | 32.36 | 35.63 |
| HiNeRV | 12.82M | 718G | 5.5/19.9 | **34.66** | **41.83** | **39.95** | **39.01** | **37.32** | **38.19** | **35.20** | **38.02** |

and decoding speeds of HiNeRV are slower than that of other methods, HiNeRV achieves a higher overall PSNR figure with fewer MACs. Further optimization may help reduce its latency. In the *Supplementary Material*, we conducted experiments with faster variants of HiNeRV, demonstrating that HiNeRV can achieve a satisfactory trade-off among latency, model size and reconstruction quality simultaneously.

Figure 1 (right) shows the performance of HiNeRV, NeRV and HNeRV (at scale S) in terms of the reconstruction quality with various epochs of training. We observe that our model with 37 epochs achieves similar reconstruction quality of HNeRV with 300 epochs on both datasets.

## 4.2 Video compression

To evaluate video compression performance, we compared HiNeRV with two INR-based models: NeRV [11] and HNeRV [10]; with two conventional codecs: HEVC/H.265 HM 18.0 (Main Profile with *Random Access*) [39, 40] and x265 (*veryslow* preset with B frames) [2]; and with two state-of-the-art learning-based codecs: DCVC [26], DCVC-HEM [27] and VCT [35]. These were all compared using two test databases: UVG [36] and MCL-JCV [51]. Unlike previous work [11, 5], which concatenates the videos and uses a single network to compress multiple videos, we train all the models for each video separately. For the training of NeRV, HNeRV and HiNeRV, we use the configurations described in Section 4.1, but with two more scales, namely XL and XXL, for encoding videos with the highest rate (See the *Supplementary Material*). We apply pruning and quantization as described in Section 3.6. In particular, we prune these three models to remove 15% of their weights and fine-tune the models for another 60 epochs. These models are further optimized with Quant-Noise [46] with 90% noise ratio for 30 epochs. Here we use the same learning rate scheduling for fine-tuning, but employ 10% of the original learning rate in the optimization with Quant-Noise. To obtain the actual rate, we perform arithmetic entropy coding [34] and combine all essential information including the pruning masks and the quantization parameters into bitstreams.

Figure 3 reports the overall rate quality performance on the UVG [36] and the MCL-JCV [51] datasets. Table 3 summarizes the average Bjøntegaard Delta (BD) rate results for both databases. All results show that HiNeRV offers competitive coding efficiency compared to most conventional codecs

Table 3: BD-Rate (Measured in PSNR/MS-SSIM) results on the UVG [36] and MCL-JCV [51] datasets.

| Dataset | Metric | x265 (*veryslow*) | HM (*RA*) | DCVC | DCVC-HEM | VCT | NeRV | HNeRV |
|---------|--------|-------------------|-----------|-------|----------|-----|------|-------|
| UVG | PSNR | -38.66% | 7.54% | -43.44% | 25.23% | -34.28% | -74.12% | -72.29% |
| | MS-SSIM | -62.70% | -41.41% | -34.50% | 49.03% | -23.69% | -73.76% | -83.86% |
| MCL-JCV | PSNR | -23.39% | 31.09% | -24.59% | 35.83% | -17.03% | -80.19% | -66.56% |
| | MS-SSIM | -44.12% | -2.65% | -17.32% | 80.73% | 12.10% | -82.28% | -79.42% |

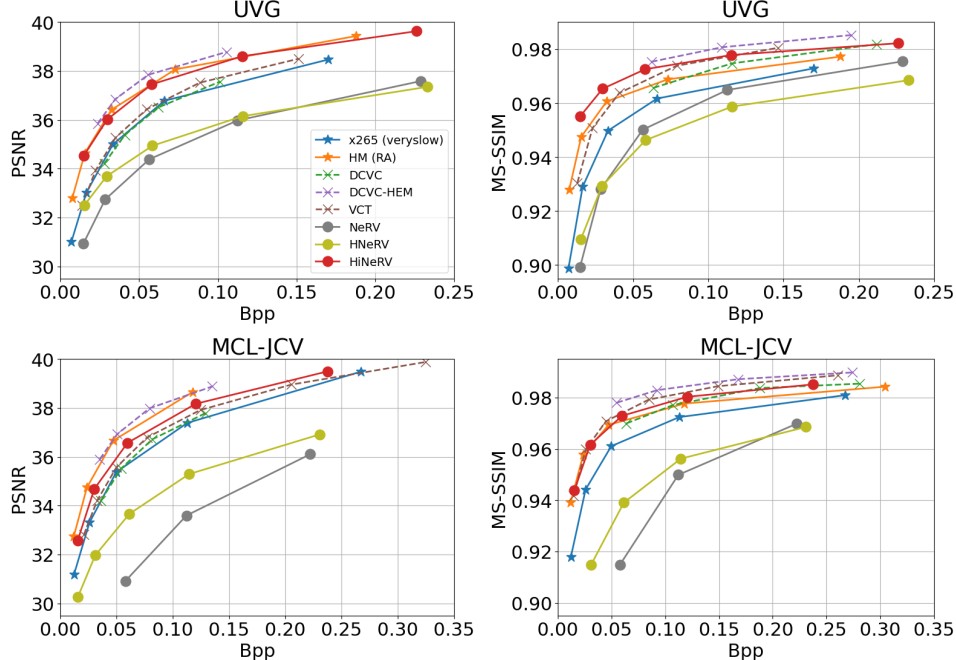

Figure 3: Video compression results on the UVG [36] and the MCL-JCV datasets [51].

and learning-based methods. This represents a significant improvement over existing NeRV-based approaches (this is also confirmed by the visual comparison with HNeRV in Figure 1). In particular, it is observed that HiNeRV outperforms x265 (*veryslow*) [2], DCVC [26] and VCT [35] based on PSNR. As far as we are aware, this is the first INR-based codec which can achieve such performance. We also observe that HiNeRV offers better performance compared to H.265 HM (*Random Access*) based on MS-SSIM. It should be noted that the results of each learning-based codec reported here are based on two model checkpoints, one for optimizing PSNR and the other for MS-SSIM, while all the results for HiNeRV are however based on the same checkpoint.

Despite the fact that HiNeRV has not been fully optimized end-to-end (entropy encoding and quantization are not optimized in the loop), it nonetheless outperforms many state-of-the-art end-to-end optimized learning-based approaches. This demonstrates the significant potential of utilizing INRs for video compression applications. In the *Supplementary Material*, comparison with additional baseline is provided.

## 4.3 Ablation study

To verify the contribution of various components in HiNeRV, we generated a number of variants of the original model, and evaluated them for video representation on the UVG dataset [36] (all the sequences were down-sampled to $1280 \times 720$ for reduction the amount of computation). For all experiments, we followed the settings in Section 4.1, and performed training targeting scale S (by adjusting the width of the network to keep the similar model sizes). All results are shown in Table 4. We also included the results of NeRV [11] and HNeRV [10] for reference.

**Bilinear interpolation with hierarchical encoding.** The contribution of bilinear interpolation with hierarchical encoding was verified by comparing it with alternative upsampling layers, sub-pixel convolutional layer [42] with $1 \times 1$ (V1) and $3 \times 3$ (V2) kernel sizes.

Table 4: Ablation studies of HiNeRV with the UVG dataset [36]. Results are in PSNR.

| Model | Size | Beauty | Bosph. | Honey. | Jockey | Ready. | Shake. | Yacht. | Avg. |
|---|---|---|---|---|---|---|---|---|---|
| NeRV | 3.20M | 34.03 | 32.77 | 39.59 | 30.39 | 23.88 | 33.85 | 26.88 | 31.63 |
| HNeRV | 3.22M | 35.04 | 35.72 | 41.11 | 32.20 | 25.88 | 35.75 | 29.69 | 33.63 |
| HiNeRV | 3.17M | 35.67 | **39.37** | 41.61 | **36.94** | **31.98** | 36.74 | **31.57** | **36.27** |
| (V1) w/ Sub-Conv1x1 | 3.16M | 35.28 | 36.63 | 41.58 | 34.64 | 29.12 | 36.31 | 29.91 | 34.78 |
| (V2) w/ Sub-Conv3x3 | 3.15M | 34.96 | 35.35 | 41.14 | 32.80 | 27.18 | 35.34 | 29.14 | 33.70 |
| (V3) w/o Encoding | 3.17M | 35.64 | 39.18 | 41.58 | 36.16 | 30.92 | 36.68 | 31.50 | 35.95 |
| (V4) w/ Fourier enc. | 3.17M | 35.62 | 39.07 | 41.59 | 36.00 | 30.91 | 36.81 | 31.47 | 35.92 |
| (V5) w/ Fourier (local) enc. | 3.17M | 35.59 | 38.99 | 41.54 | 35.77 | 30.61 | 36.57 | 31.30 | 35.77 |
| (V6) w/ Grid (local) enc. | 3.19M | 35.65 | 39.26 | 41.58 | 36.17 | 30.93 | 36.72 | 31.55 | 35.98 |
| (V7) w/ MLP | 3.19M | 35.10 | 37.17 | 41.35 | 34.77 | 29.10 | 35.58 | 29.76 | 34.69 |
| (V8) w/ Conv3x3 | 3.17M | 35.35 | 37.86 | 41.37 | 35.13 | 29.70 | 36.10 | 30.31 | 35.12 |
| (V9) w/ Frame-wise | 3.17M | **35.68** | 39.22 | 41.54 | 36.69 | 31.49 | 36.54 | 31.54 | 36.10 |
| (V10) w/ Patch-wise | 3.17M | 35.46 | 38.30 | 41.55 | 35.04 | 30.06 | 36.51 | 30.77 | 35.38 |
| (V11) w/ Nearest Neighbor | 3.17M | 35.60 | 39.12 | **41.64** | 36.52 | 31.51 | **36.82** | 31.33 | 36.08 |

**Upsampling encodings.** Four variants are trained to confirm the effectiveness of the upsampling encodings including (V3) w/o encoding; (V4) with the Fourier encoding [37]; (V5) using Fourier encoding with local coordinates (computed in the hierarchical encoding); (V6) using the grid-based encoding with local coordinates, i.e. the hierarchical encoding without the temporal dimension.

**ConvNeXt block.** The employed ConvNeXt block [31] has been compared with (V7) the MLP block in transformer [50]; (V8) the block containing two convolutional layers with , where we use $3 \times 3$ and $1 \times 1$ kernel size to keep the receptive field consistent with the ConvNeXt block used in our paper.

**Unified representations.** V9 and V10 have been generated for frame- and patch-wise configurations.

**Interpolation methods.** V11 replaces the bilinear interpolation by the nearest neighbor interpolation.

The ablation study results are presented in Table. 4 which shows that the full HiNeRV model outperforms all alternatives (V1-V11) on the UVG dataset in terms of the overall reconstruction quality. This confirms the contribution of each primary component of the design. More discussion regarding the results can be found in the *Supplementary Material*.

# 5 Conclusion

In this paper, a new neural representation model, HiNeRV, has been proposed for video compression, which exhibits superior coding performance over many conventional and learning-based video codecs (including those based on INRs). The improvements demonstrated are associated with new innovations including bilinear interpolation based hierarchical encoding, a unified representation and a refined model compression pipeline.

Despite the fact that HiNeRV has not been fully optimized end-to-end (entropy encoding and quantization are not optimized in the loop), it nonetheless achieves comparable performance to state-of-the-art end-to-end optimized learning-based approaches, with significant improvement over existing NeRV-based algorithms. This demonstrates the great potential of utilizing INRs for video compression applications. For example, this is the first INR-based video codec which can outperform HEVC HM Random Access mode based on MS-SSIM.

Future work should focus on incorporation of entropy coding and quantization to achieve full end-to-end optimization.

# Acknowledgment

This work was jointly funded by UK EPSRC (iCASE Awards), BT, the UKRI MyWorld Strength in Places Programme and the University of Bristol. We also thank the support by the Advanced Computing Research Centre, University of Bristol, for providing the computational facilities.

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
