# HiNeRV: Video Compression with Hierarchical Encoding-based Neural Representation
## *Supplementary Material*

**Ho Man Kwan†, Ge Gao†, Fan Zhang†, Andrew Gower‡, David Bull†**
† Visual Information Lab, University of Bristol, UK
‡ Immersive Content & Comms Research, BT, UK
{hm.kwan, ge1.gao, fan.zhang, dave.bull}@bristol.ac.uk,
andrew.p.gower@bt.com

## A   Implementation details

In this work, we implemented the proposed HiNeRV and other NeRV-based models [7, 12, 4, 9, 6] using the PyTorch [18] framework and the PyTorch Image Models library [26]. We used torchac [14] for performing arithmetic coding.

For training NeRV, E-NeRV, HNeRV and FFNeRV, we adopted their original implementations, while we re-implemented PS-NeRV based on its original description due to the unavailability of the source code. All these models were trained on GPUs with half-precision [17]. It was observed that training HNeRV with half-precision results in inferior results for some sequences in the MCL-JCV dataset, so we reported its results with full precision.

For learning-based methods, we evaluated the performance of DCVC [10] and DCVC-HEM [11] using the implementations and pre-trained models created by the authors and reported the performance of VCT [15] and B-EPIC [19] using the RD data provided in the corresponding GitHub repository/paper.

In our experiments, we estimated the complexity (MACs) of different models using the DeepSpeed library [1]. It is noted that in the original implementations of some benchmarked methods [7, 12, 9], the provided configurations adopt a large number (e.g. 96) as the minimum width of the networks, resulting in high computational complexity (due to the high resolution feature maps generated) and thus large MACs figures despite of their relatively small model sizes.

For conventional codecs, we performed experiments with multiple QP values to obtain the results at different rates. Specifically, we used QP values 17/22/27/32/37/42 and 12/17/22/27/32/37 for x265 [3] and HM [20], respectively.

## B   Comparison to other learning-based codecs

Comparing with learning-based codecs, the main advantage of an INR-based model is the decoding speed [7]. Although HiNeRV offers superior compression and representation performance compared to existing INR-based approaches (as demonstrated in the main paper), due to the more sophisticated structure employed, its encoding and decoding speeds are also slower than other INR-based methods. However, it should be noted that, when compared with other learning-based codecs with state-of-the-art compression performance, HiNeRV still exhibits a much faster decoding speed. For example, the decoding speeds of DCVC [10] and DCVC-HEM [11] with 1080p videos are 0.03/1.90 FPS (reported in [11]), while HiNeRV (scale L) can obtain 10.9 FPS on the same GPU. Moreover, HiNeRV has a similar complexity (in MACs) when compared with faster INR-based model, i.e., HNeRV [6], and further optimization is expected to reduce the actual latency.

37th Conference on Neural Information Processing Systems (NeurIPS 2023).

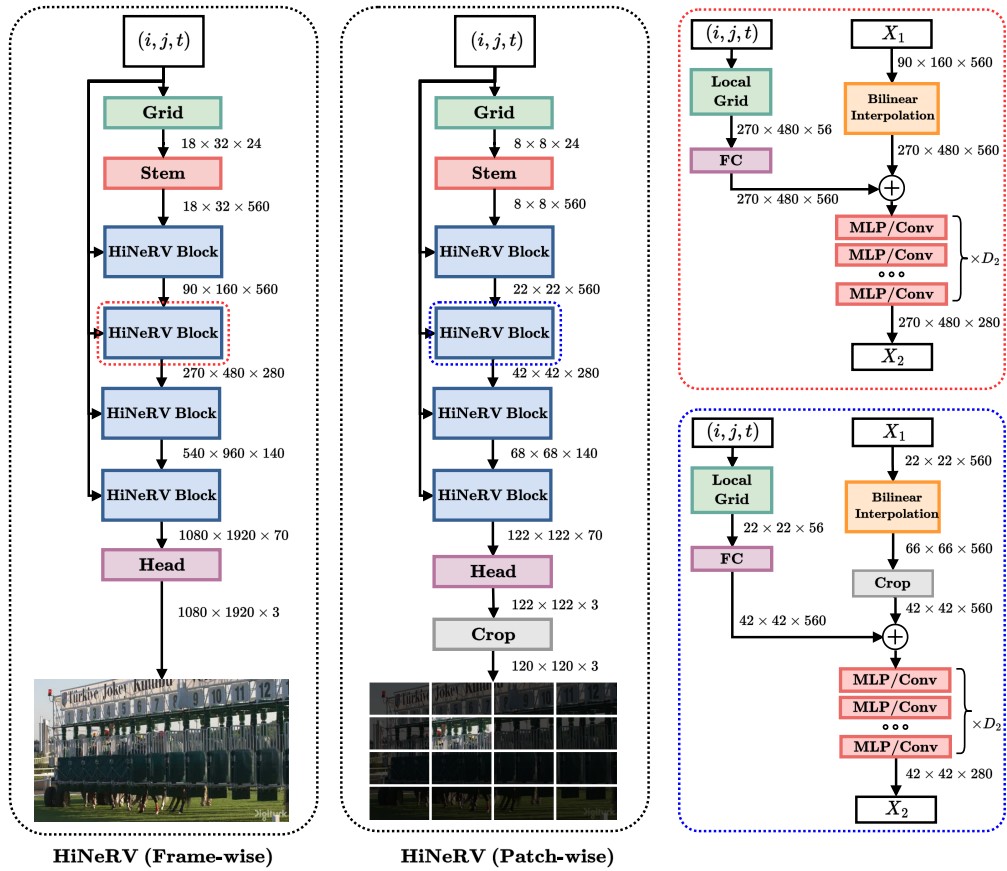

Figure 1: Illustration of the proposed HiNeRV models employing frame-based or patch-based representation.

Encoding speed is a common issue affecting all NeRV-based approaches, which require model training as part of video encoding. With the experiment settings adopted in Section 4.2, encoding a 1080p video with 600 frames by HiNeRV takes around 6.5/7.7/11.9 hours with scale S/M/L, respectively. However, as pointed out in the paper (Section 4.1), it is possible to reduce the encoding time while still obtaining superior reconstruction quality.

We also noticed that when comparing with learning-based codecs, HiNeRV performs better with the UVG dataset [16] than with the MCL-JCV [25] dataset. This could be due to the fact that the UVG dataset contains a larger number of frames per sequence, where the INR based method is able to take advantage of it. Future work could also investigate this limitation.

## C    Details for unifying frame-wise and patch-wise representations (Sec. 3.5)

When the patch-wise configuration is employed (without padding), operations such as convolution and bilinear interpolation will produce different results to those based on the frame-wise configuration due to boundary effects. For HiNeRV, we perform computation with overlapped patches in the feature map space to remove the negative effect of the boundary pixels, where the amount of overlapped pixels increase with the number of convolutional/interpolation layers and kernel sizes. It is noted that, due to the use of convolutional layers, masking is also required to set the pixel values outside of the boundary (related to frame) to zero, in order to match the behavior of convolutional layers in deep learning frameworks with the commonly used 'zero padding' setting.

Fig. 1 provides an illustrative comparison between the frame-wise and patch-wise configurations used in HiNeRV. Here we take HiNeRV (scale L) as an example (see Section D for the detailed configurations). First, giving the input patch coordinate $(i, j, t)$ (for the frame-wise configuration, $i = j = 0$), the base encoding is interpolated from the feature grid in $\gamma_{base}$. In frame-wise mode,

Table 1: HiNeRV configurations for the UVG [16] and MCL-JCV [25] datasets.

| Size | $D_n$ | $S_n$ | $C_0$ | $(T_{grid}, H_{grid}, W_{grid}, C_{grid})$ | $L_{grid}$ | $(T_{local}, C_{local})$ | $L_{local}$ |
|------|-------|-------|-------|---------------------------------------------|------------|--------------------------|-------------|
| XXS | $(3, 3, 3, 1)$ | $(5, 3, 2, 2)$ | 136 | MCL-JCV - $(40, 18, 32, 2)$ | 2 | MCL-JCV - $(T, 4)$ | 3 |
| XS | $(3, 3, 3, 1)$ | $(5, 3, 2, 2)$ | 196 | MCL-JCV - $(40, 18, 32, 4)$ | 2 | MCL-JCV - $(T, 8)$ | 3 |
| S | $(3, 3, 3, 1)$ | $(5, 3, 2, 2)$ | 280 | MCL-JCV - $(40, 18, 32, 8)$ 
 UVG - $(150, 18, 32, 2)$ | 2 | MCL-JCV - $(T, 16)$ 
 UVG - $(T, 4)$ | 3 |
| M | $(3, 3, 3, 1)$ | $(5, 3, 2, 2)$ | 400 | MCL-JCV - $(40, 18, 32, 16)$ 
 UVG - $(150, 18, 32, 4)$ | 2 | MCL-JCV - $(T, 32)$ 
 UVG - $(T, 8)$ | 3 |
| L | $(3, 3, 3, 1)$ | $(5, 3, 2, 2)$ | 560 | MCL-JCV - $(40, 18, 32, 32)$ 
 UVG - $(150, 18, 32, 8)$ | 2 | MCL-JCV - $(T, 64)$ 
 UVG - $(T, 16)$ | 3 |
| XL | $(4, 4, 4, 1)$ | $(5, 3, 2, 2)$ | 688 | UVG - $(150, 18, 32, 16)$ | 2 | UVG - $(T, 32)$ | 3 |
| XXL | $(5, 5, 5, 1)$ | $(5, 3, 2, 2)$ | 864 | UVG - $(150, 18, 32, 32)$ | 2 | UVG - $(T, 64)$ | 3 |

$T$: the number of video frames

Table 2: Video representation results on the Bunny dataset [2] (for XXS/XS/S scales).

| Model | Size | MS-SSIM |
|-------|------|---------|
| NeRV | 0.83M/1.64M/3.20M | 0.8441/0.9189/0.9623 |
| E-NeRV | 0.88M/1.65M/3.31M | 0.9392/0.9678/0.9873 |
| PS-NeRV | 0.90M/1.68M/3.35M | 0.9478/0.9632/0.9769 |
| HNeRV | 0.82M/1.66M/3.28M | 0.9558/0.9773/0.9892 |
| FFNeRV | 0.91M/1.66M/3.19M | 0.9559/0.9773/0.9891 |
| HiNeRV | 0.77M/1.59M/3.25M | **0.9861/0.9922/0.9955** |

Table 3: Video representation results on the UVG dataset [16] (for S/M/L scales). Results are in MS-SSIM.

| Model | Size | Beauty | Bosph. | Honey. | Jockey | Ready. | Shake. | Yacht. | Avg. |
|-------|------|--------|--------|--------|--------|--------|--------|--------|------|
| NeRV | 3.31M | 0.8862 | 0.9214 | 0.9826 | 0.8871 | 0.8303 | 0.9336 | 0.8539 | 0.8993 |
| E-NeRV | 3.29M | 0.8876 | 0.9341 | 0.9842 | 0.8627 | 0.8523 | 0.9407 | 0.8665 | 0.9040 |
| PS-NeRV | 3.24M | 0.8781 | 0.9105 | 0.9804 | 0.8397 | 0.7807 | 0.9409 | 0.8381 | 0.8823 |
| HNeRV | 3.26M | 0.8941 | 0.9503 | 0.9846 | 0.8775 | 0.8402 | 0.9473 | 0.8865 | 0.9115 |
| FFNeRV | 3.40M | 0.8977 | 0.9590 | 0.9846 | 0.8804 | 0.8564 | 0.9473 | 0.8904 | 0.9165 |
| HiNeRV | 3.19M | **0.9067** | **0.9843** | **0.9857** | **0.9571** | **0.9672** | **0.9648** | **0.9489** | **0.9592** |
| NeRV | 6.53M | 0.8996 | 0.9542 | 0.9843 | 0.9247 | 0.8909 | 0.9454 | 0.8990 | 0.9283 |
| E-NeRV | 6.54M | 0.9015 | 0.9618 | 0.9854 | 0.9111 | 0.9061 | 0.9593 | 0.9098 | 0.9336 |
| PS-NeRV | 6.57M | 0.8948 | 0.9464 | 0.9842 | 0.8478 | 0.8478 | 0.9574 | 0.8853 | 0.9144 |
| HNeRV | 6.40M | 0.9014 | 0.9634 | 0.9853 | 0.9095 | 0.8860 | 0.9607 | 0.9140 | 0.9315 |
| FFNeRV | 6.44M | 0.9036 | 0.9652 | 0.9855 | 0.9375 | 0.9285 | 0.9628 | 0.9238 | 0.9438 |
| HiNeRV | 6.49M | **0.9162** | **0.9886** | **0.9862** | **0.9683** | **0.9814** | **0.9744** | **0.9675** | **0.9689** |
| NeRV | 13.01M | 0.9103 | 0.9717 | 0.9854 | 0.9508 | 0.9363 | 0.9650 | 0.9365 | 0.9509 |
| E-NeRV | 13.02M | 0.9075 | 0.9745 | 0.9858 | 0.9434 | 0.9407 | 0.9726 | 0.9391 | 0.9519 |
| PS-NeRV | 13.07M | 0.9016 | 0.9651 | 0.9853 | 0.9111 | 0.9713 | 0.9713 | 0.9221 | 0.9395 |
| HNeRV | 12.87M | 0.9066 | 0.9739 | 0.9857 | 0.9369 | 0.9261 | 0.9721 | 0.9389 | 0.9486 |
| FFNeRV | 12.66M | 0.9144 | 0.9793 | 0.9860 | 0.9589 | 0.9587 | 0.9753 | 0.9532 | 0.9608 |
| HiNeRV | 12.82M | **0.9277** | **0.9911** | **0.9876** | **0.9739** | **0.9885** | **0.9809** | **0.9798** | **0.9756** |

the base encoding has a spatial size of $18 \times 32$. For the patch-wise mode, if padding is not used, the base encoding size will be $2 \times 2$, as we segment the frames into $9 \times 16$ patches, and compute one patch in each forward pass. With the padding (3 pixels on each side, in this case), the output becomes $8 \times 8$. Each HiNeRV block performs upsampling and cropping which adapt with the padding size. For example, in the second HiNeRV block for both configurations (highlighted in the figure), bilinear interpolation is used to upsample the feature maps by $3\times$. In the case of the patch-wise mode, center cropping is applied after upsampling, where the crop size depends on the padding size in the new resolution. Similarly, cropping is also performed after the head layer.

It should be noted that, although padding will introduce an additional computational overhead, the above approach can improve the performance of HiNeRV as shown in the main paper. This overhead can be further reduced by increasing the patch size if the computational complexity is a concern. It becomes zero if a frame-wise configuration is employed in the inference.

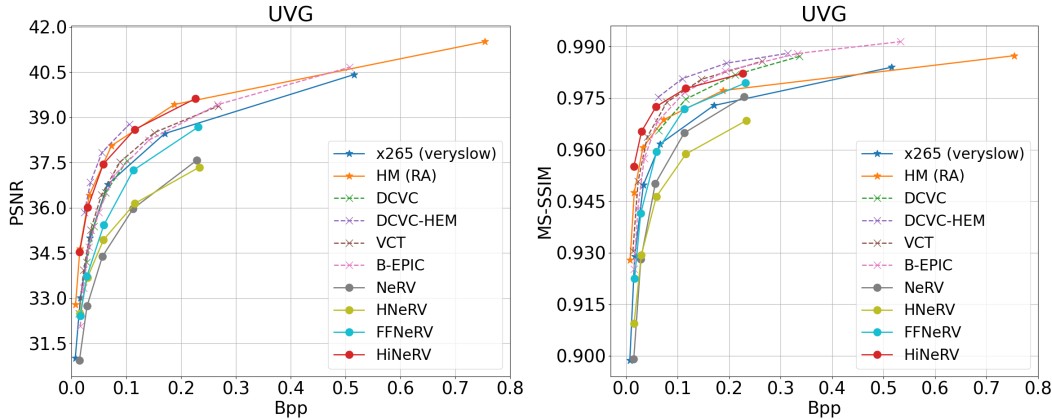

Figure 2: Video compression results on the UVG datasets [16].

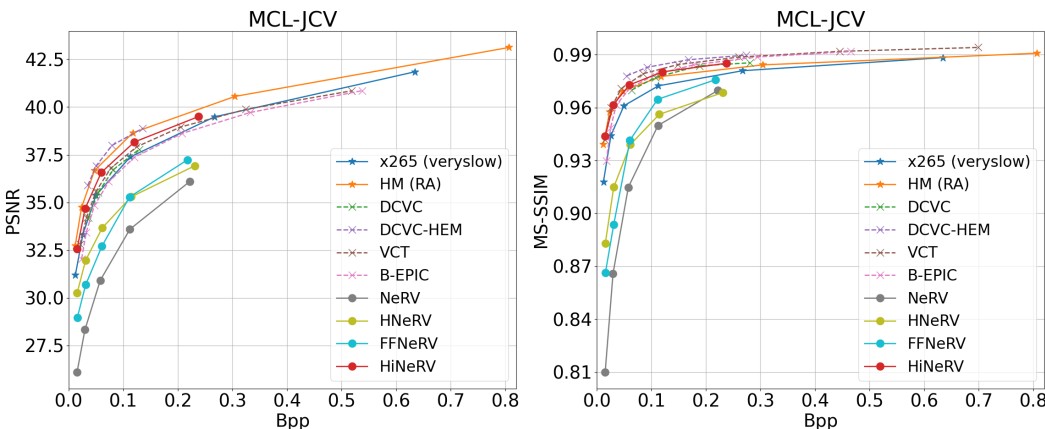

Figure 3: Video compression results on the MCL-JCV datasets [25].

# D  Configurations for HiNeRV

In the proposed approach, we use 4 HiNeRV blocks, i.e. $N = 4$, with a reduction factor $R = 2.0$. The detailed configurations of HiNeRV for the evaluation on the UVG [16] and MCL-JCV [25] datasets are given in Table 1. The settings for the Bunny dataset [2], are identical to those for MCL-JCV, except that the strides $S_n$ are changed to $(5, 2, 2, 2)$ to match the spatial resolution. The employed network architecture adopts ConvNeXt [13] as the default network block, and we use a $3 \times 3$ kernel size for all convolutional layers, and an expansion ratio of $4$ for all ConvNeXt blocks except for the last one, which has a ratio of $1$, to reduce the computational complexity. We did not fine-tune the configurations thoroughly, but we noticed that HiNeRV is relatively robust to parameter changes, e.g. adjusting the depth and width, given the same number of parameters. For scaling HiNeRV, we primarily increase the width instead of the depth to avoid large padding sizes, as the padding sizes are scaled with both the network depth and the kernel size of convolutional layers.

For patch-wise computation (with or without padding), we use a patch size $M = 80/120$ for $1280 \times 720/1920 \times 1080$ outputs. We configure the padding size of the stem layer and four HiNeRV blocks independently, where we use a padding size $(3, 6, 6, 4, 1)$ for HiNeRV-XXS/S/M/L, $(3, 7, 7, 5, 1)$ for HiNeRV-XL and $(4, 9, 9, 6, 1)$ for HiNeRV-XXL, respectively. For a convolutional layer with a kernel size $K$, the padding required can be computed by $\lceil \frac{K-1}{2} \rceil$. For the upsampling layers, the padding can by obtained by calculating the pixel position for interpolation. The total padding required is accumulated in a top-down manner.

Table 4: Comparison between different pruning configurations. Results are based on the UVG dataset [16], measured in average PSNR.

| | Sparsity | | | | |
|---|---|---|---|---|---|
| Method | 0.1500 | 0.2775 | 0.3859 | 0.4780 | 0.5563 |
| Original | 36.08 | 31.97 | 31.63 | 31.20 | 30.72 |
| Adaptive (ours) | **36.13** | **35.79** | **35.40** | **34.97** | **34.50** |
| Head excluded | 36.08 | 35.65 | 35.22 | 34.69 | 34.09 |

Table 5: Comparison between different quantization configurations. Results are based on the UVG dataset [16], measured in average PSNR.

| | Bitwidth | | | |
|---|---|---|---|---|
| Configuration | 8 | 7 | 6 | 5 |
| None | 36.14 | 35.75 | 34.59 | 32.08 |
| QAT | 36.18 | 36.02 | 35.50 | 34.64 |
| Quant-Noise (w/ STE) | 36.18 | 36.04 | 35.55 | 34.68 |
| Quant-Noise (w/o STE) | **36.20** | **36.12** | **35.86** | **35.16** |

# E   Additional results

## E.1   Video representation

Additional MS-SSIM results for the video representation task (Section 4.1 in the main paper) on the Bunny [2] and the UVG [16] datasets are summarized in Table 2 and Table 3, respectively.

## E.2   Video compression

For the video compression task (Section 4.2 in the main paper), additional results have been provided in Figure 2 and 3 for the UVG [16] and the MCL-JCV [25] datasets respectively. Here two more learning-based codecs are included for benchmarking including FFNeRV [9] and B-EPIC [19].

## E.3   Ablation study on the refined training pipeline

Alongside the ablation study in the main paper, additional experiments were also performed to verify the effectiveness of the refined training pipeline (Section 3.6). The experiments in this sub-section adopt the same settings as in Section 4.3.

Firstly, we compare the different pruning techniques including: (i) the standard pruning technique utilized in the original model compression pipeline for NeRV [7] and (ii) with the proposed adaptive pruning in this paper. Empirically, we found that (i) can lead to a larger portion of the head weights being pruned, with the network performance unable to recover even with fine-tuning. Hence, we consider an alternative, (iii), using the standard pruning technique with the head weights excluded. To obtain networks with different pruning ratios, we perform multiple iterations of pruning, with each removing 15% of the weights, followed by fine-tuning (60 rounds). The results in Table 4 show that the proposed adaptive pruning method (ii) is superior to the other two methods, especially when the pruning ratio is larger.

We also compared the performance of models with quantization, where they are trained (i) without quantization-aware training (QAT) [8], (ii) with QAT, where Straight Through Estimator (STE) [5] is being used, (iii) with Quant-Noise [22], where STE is being used to compute the gradient for the quantized weights, or (iv) with Quant-Noise [22], where STE is not used, i.e., our choice in the main paper. The results for 5/6/7 and 8 bit quantisation are provided. For training with QAT and Quant-Noise, we fine-tune the models over 30 epochs. The results in Table 5 show that, by using Quant-Noise (w/o STE) with 6 bits, we can obtain a relatively good performance, while 6-bit quantization can provide up-to 25% bitrate saving compared with the commonly used 8-bit alternative [7, 12, 4, 9, 6].

Table 6: Video representation results on the Bunny dataset [2] for the faster HiNeRV variants (for XXS/XS/S scales).

| Model | Size | MACs | Encoding FPS | Decoding FPS | PSNR |
|---|---|---|---|---|---|
| NeRV | 0.83M/1.64M/3.20M | 25G/57G/101G | **131.9/93.6/81.8** | 308.5/229.1/**202.3** | 26.82/29.61/32.56 |
| HNeRV | 0.82M/1.66M/3.28M | 23G/48G/94G | 100.0/80.8/64.2 | **317.4/251.6**/192.5 | 31.08/33.68/36.95 |
| FFNeRV | 0.91M/1.66M/3.19M | 26G/58G/102G | 62.1/51.5/47.9 | 108.4/95.2/90.5 | 30.37/33.83/37.01 |
| HiNeRV | 0.77M/1.59M/3.25M | 23G/47G/96G | 37.6/27.7/20.0 | 132.1/103.9/76.7 | 36.37/**38.94/41.14** |
| HiNeRV-A | 0.76M/1.57M/3.22M | 18G/37G/76G | 41.0/30.8/21.7 | 139.2/109.9/81.9 | **36.50**/38.84/40.87 |
| HiNeRV-B | 0.79M/1.57M/3.25M | 12G/24G/48G | 75.0/62.9/44.5 | 196.6/162.4/114.9 | 35.31/37.57/39.78 |

Table 7: Video representation results with the UVG dataset [16] for the faster HiNeRV variants (for S/M/L scales). Results are in PSNR. FPS is the encoding/decoding rate.

| Model | Size | MACs | FPS | Beauty | Bosph. | Honey. | Jockey | Ready. | Shake. | Yacht. | Avg. |
|---|---|---|---|---|---|---|---|---|---|---|---|
| NeRV | 3.31M | 227G | **32.4**/90.0 | 32.83 | 32.20 | 38.15 | 30.30 | 23.62 | 33.24 | 26.43 | 30.97 |
| HNeRV | 3.26M | 175G | 24.6/**93.4** | 33.56 | 35.03 | 39.28 | 31.58 | 25.45 | 34.89 | 28.98 | 32.68 |
| FFNeRV | 3.40M | 228G | 19.0/49.3 | 33.57 | 35.03 | 38.95 | 31.57 | 25.92 | 34.41 | 28.99 | 32.63 |
| HiNeRV | 3.19M | 181G | 10.1/35.5 | **34.08** | **38.68** | **39.71** | 36.10 | **31.53** | **35.85** | **30.95** | **35.27** |
| HiNeRV-A | 3.22M | 122G | 12.0/40.3 | 34.06 | 38.31 | 39.65 | **36.27** | 31.08 | 35.68 | 30.71 | 35.11 |
| HiNeRV-B | 3.22M | 78G | 22.4/56.9 | 33.81 | 37.07 | 39.42 | 35.27 | 29.43 | 34.90 | 29.60 | 34.21 |
| NeRV | 6.53M | 228G | **32.0/90.1** | 33.67 | 34.83 | 39.00 | 33.34 | 26.03 | 34.39 | 28.23 | 32.78 |
| HNeRV | 6.40M | 349G | 20.1/68.5 | 33.99 | 36.45 | 39.56 | 33.56 | 27.38 | 35.93 | 30.48 | 33.91 |
| FFNeRV | 6.44M | 229G | 18.9/49.3 | 33.98 | 36.63 | 39.58 | 33.58 | 27.39 | 35.91 | 30.51 | 33.94 |
| HiNeRV | 6.49M | 368G | 8.4/29.1 | **34.33** | **40.37** | **39.81** | 37.93 | **34.54** | **37.04** | **32.94** | **36.71** |
| HiNeRV-A | 6.56M | 246G | 10.0/33.3 | 34.29 | 39.97 | 39.77 | **37.95** | 33.87 | 36.87 | 32.59 | 36.47 |
| HiNeRV-B | 6.40M | 150G | 17.8/44.4 | 34.06 | 38.68 | 39.63 | 36.90 | 31.60 | 35.95 | 31.11 | 35.42 |
| NeRV | 13.01M | 230G | **31.7/89.8** | 34.15 | 36.96 | 39.55 | 35.80 | 28.68 | 35.90 | 30.39 | 34.49 |
| HNeRV | 12.87M | 701G | 15.6/52.7 | 34.30 | 37.96 | 39.73 | 35.47 | 29.67 | 37.16 | 32.31 | 35.23 |
| FFNeRV | 12.66M | 232G | 18.4/49.3 | 34.28 | 38.48 | 39.74 | 36.72 | 30.75 | 37.08 | 32.36 | 35.63 |
| HiNeRV | 12.82M | 718G | 5.5/19.9 | **34.66** | **41.83** | **39.95** | **39.01** | 37.32 | **38.19** | **35.20** | **38.02** |
| HiNeRV-A | 12.96M | 478G | 6.7/23.1 | 34.58 | 41.36 | 39.92 | 38.96 | 36.38 | 37.99 | 34.46 | 37.66 |
| HiNeRV-B | 13.08M | 302G | 11.6/28.2 | 34.36 | 40.31 | 39.77 | 38.16 | 34.04 | 37.20 | 33.10 | 36.71 |

## E.4 Faster variants of HiNeRV

While HiNeRV (in the main paper) was configured to prioritize compression performance, we provide experimental results for two variants in this section associated with reduced computational cost (in MACs) and improved encoding/decoding speed, but with only a small drop in reconstruction quality.

The first variant (HiNeRV-A) is obtained by reducing the feature map size in the lower level layer of the network, which is achieved by using the strides $S_n = (5, 4, 2, 2)/(5, 4, 3, 2)$ for $1280 \times 720/1920 \times 1080$ output, and we adjusted the grid's spatial dimension to $16 \times 9$ accordingly. In the second variant (HiNeRV-B), we further reduce the number of network blocks in high level HiNeRV blocks ($D_n = (2, 1, 1, 1)$), and remove the normalization layer after upsampling to reduce the latency. We change the width to maintain the size of the network at each scale.

We compare HiNeRV-A/B with NeRV [7], HNeRV [6], FFNeRV [9] and the original HiNeRV. The results provided in Table 6 and 7 demonstrate much lower MAC figures with both variants of HiNeRV compared with the original version, while still achieving competitive performance. HiNeRV-A has reduced the number of operations by up to $1/3$, but still obtaining comparable performance with the original HiNeRV. The MACs figure of HiNeRV-A is much smaller than the HNeRV with the same scale. HiNeRV-B further reduced the number of operations and improved the FPS significantly. When comparing HiNeRV-B with HNeRV, where HNeRV requires two times larger sizes (S vs M/M vs L), the former always outperforms HNeRV in terms of the average reconstruction quality. However HiNeRV-B requires only half of the size, nearly one quarter of the number of operations, and is able to achieve faster encoding speed and around 80% of HNeRV decoding speed. It should be noted that, given the significantly smaller amount of MACs required, further optimization may result in HiNeRV-B having a quicker decoding speed than HNeRV.

## E.5 Discussion regarding the ablation studies

**Bilinear interpolation with hierarchical encoding.** Our results verified that the use of bilinear interpolation with hierarchical encoding is a better choice than the use of convolutional layers. When

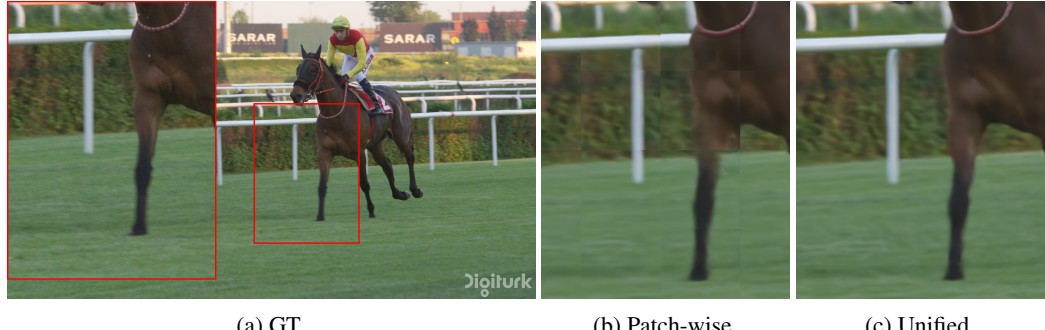

    (a) GT.                                  (b) Patch-wise.          (c) Unified.

Figure 4: Comparison between the output of the patch-wise representation and unified representation on the Jockey sequence from the UVG dataset [16].

replacing the upsampling layer with $3 \times 3$ sub-pixel convolutional layers [21], HiNeRV can only match HNeRV [6] by the average performance. When using the $1 \times 1$ variant, the performance is improvement significantly, but still far behind the HiNeRV with the proposed upsampling layer.

**Upsampling encodings.** The results suggest that the proposed hierarchical encoding provides superior performance. When Fourier encoding or the grid-based encoding (without temporal dimension) is being used, the network performance is just close to the one without encoding. The use of the proposed hierarchical encoding provides largest boost for some hard sequences, i.e., the Jockey and ReadySetGo sequences [16], which contain both fast motion and high contrast content. The improvements are 0.78/1.06 dB PSNR, respectively.

It is worth note that, there is one exceptional case where the variant with the Fourier encoding outperformed HiNeRV with the hierarchical encoding, which is the ShakeNDry sequence [16]. This could be caused by the fact that the sequence contains content with a lot of small particle motion, and these features could be better represented by Fourier-based encoding, which contains absolute positional information.

**ConvNeXt block.** The results show that the HiNeRV with ConNeXT block [13] outperforms the variants with either the MLP block [24] or the normal convolutional block. We also observed that the variant with MLP block can still outperform NeRV [7] and HNeRV [6] significantly, where previous work has suggested that NeRV perform better [7]. We assert that this is mainly due to the use of hierarchical encoding, and we performed additional experiments to validate this. The average PSNR of MLP-based HiNeRV dropped from 34.69 dB to 32.16 dB after removing the hierarchical encoding, and the performance is no longer better than NeRV and HNeRV. This suggests that hierarchical encoding could be useful for future work in MLP-based neural representations.

**Unified representations.** In most cases, training HiNeRV with the frame-wise or patch-wise configuration reduces its coding performance. This reduction is most significant for the challenging sequences, i.e., the Jockey and ReadySetGo sequences [16], where the decrease is 0.25/0.49 dB and 1.90/1.92 dB for the frame-wise and patch-wise configurations, respectively.

**Interpolation methods.** By replacing Bilinear interpolation with the Nearest Neighbor variant, the performance of HiNeRV is only reduced by a small amount. This suggests that the main improvement in HiNeRV is not achieved through Bilinear interpolation but rather through the overall design and the parameter efficiency of using interpolation for upsampling.

### E.6   Qualitative comparison between different types of representation

A comparison between the outputs of patch-wise and the proposed unified representation is shown in Figure 4. These results are obtained from HiNeRV using the settings in Section 4.3 (main paper). While both of these are performed in a patch-wise fashion, the unified representation can be applied on either overlapped patches or whole video frames. The output from the patch-wise representation exhibits noticeable artifacts around the boundaries on the patches, which do not appear in the output of the proposed representation.

### E.7 Qualitative comparison between NeRV, HNeRV and HiNeRV

Additional visual comparisons between the outputs from NeRV [7], HNeRV [6] and HiNeRV are given in Figure 5 - 8. These are obtained using the models presented in Section 4.2, where the HiNeRV model has approximately half of the size of those for NeRV and HNeRV. Despise halving the bitrate, the output from HiNeRV is still noticeably better, with more detail from the original video frames preserved.

## F   Limitations

As mentioned in Section B, one main limitation of HiNeRV is the slow encoding speed. It takes multiple hours for compressing a short video sequence, if a high encoding quality is needed. This is also a problem of all existing INR-based methods. For future work, investigations should be conducted to further speed up the training process. For example, meta-learning has been utilized to speed-up the training of neural representations [23] in other domains. This has the potential to benefit the representation for videos as well.

Comparing with other INR-based methods, the relatively complex HiNeRV network structure also leads to a larger memory footprint. However, the proposed unified representation allows operations in patches, which can significantly reduce the required memory and improve the parallelism when performing training and inference.

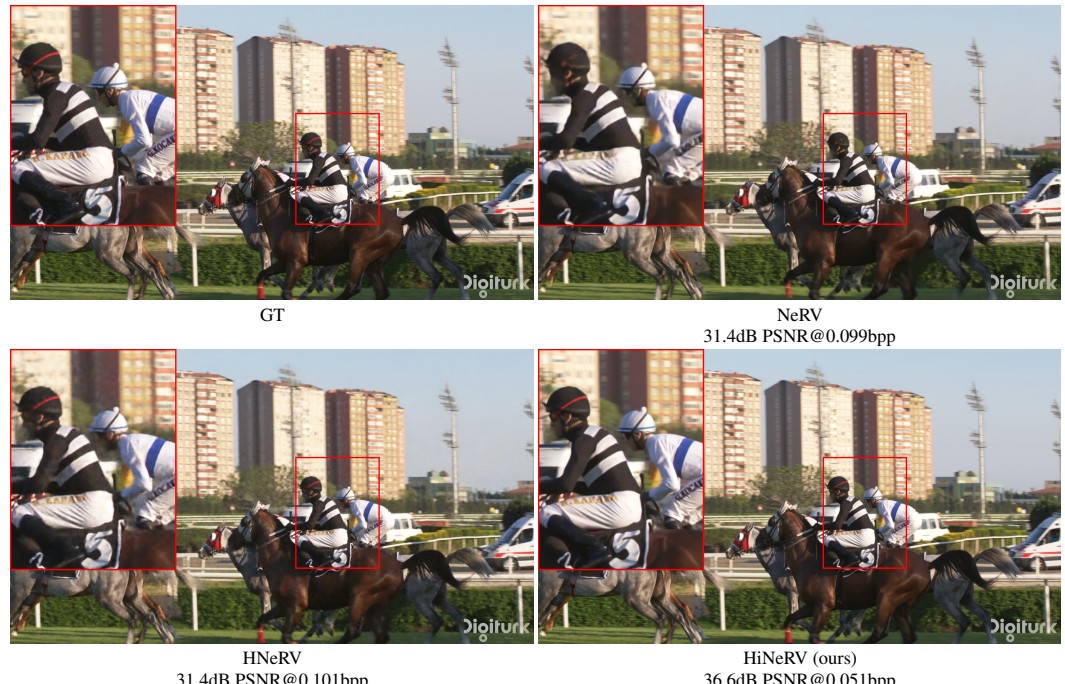

Figure 5: Comparison between the output of NeRV, HNeRV and HiNeRV with the ReadySetGo sequence from the UVG dataset [16].

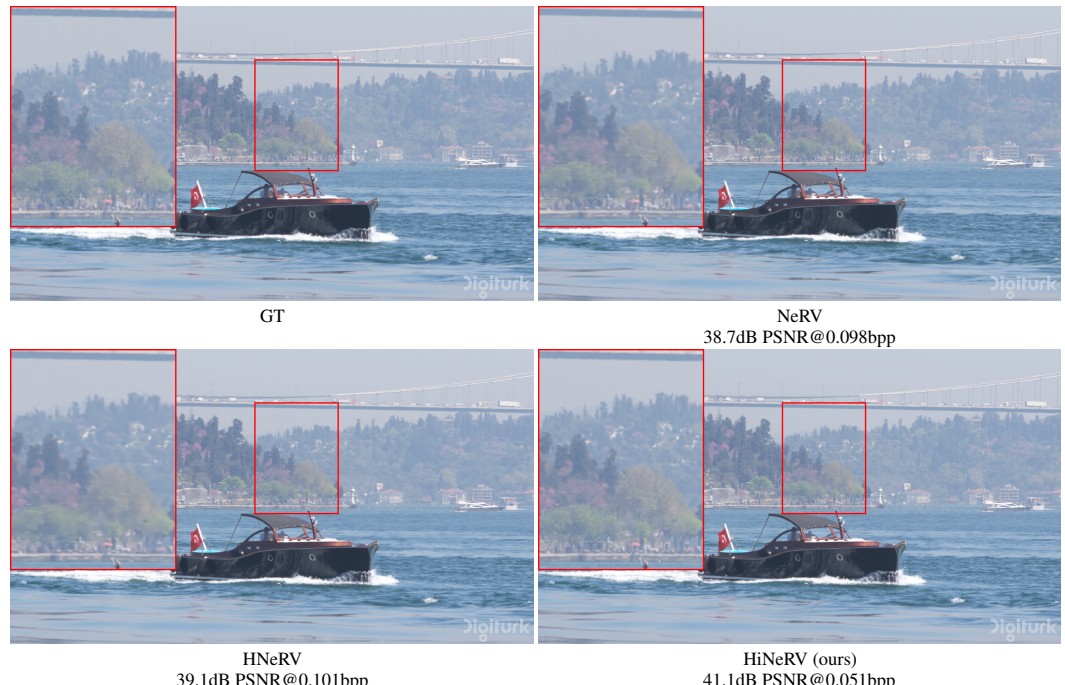

Figure 6: Comparison between the output of NeRV, HNeRV and HiNeRV with the Bosphorus sequence from the UVG dataset [16].

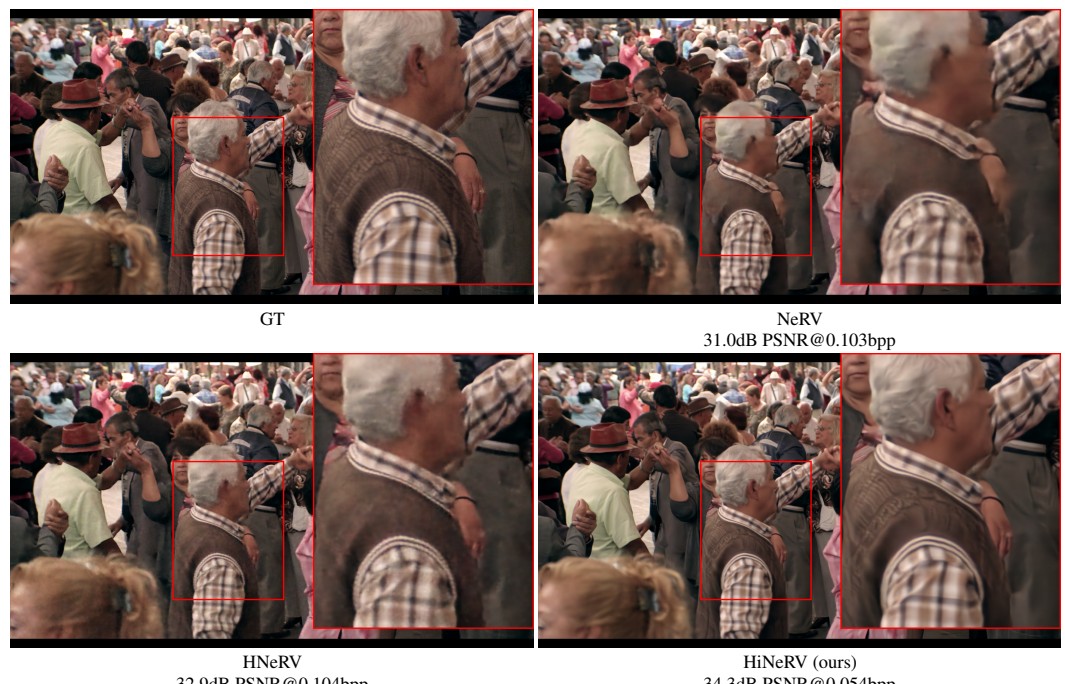

Figure 7: Comparison between the output of NeRV, HNeRV and HiNeRV with the videoSRC14 sequence from the MCL-JCV dataset [25].

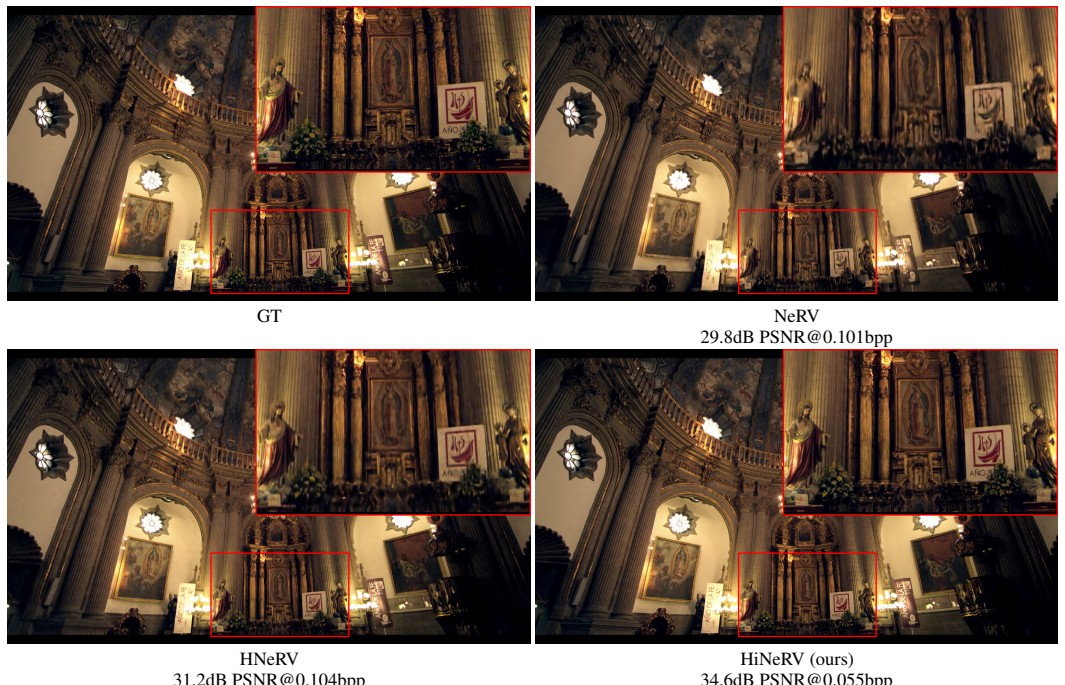

Figure 8: Comparison between the output of NeRV, HNeRV and HiNeRV with the videoSRC15 sequence from the MCL-JCV dataset [25].