# OpenReview forum: "HiNeRV: Video Compression with Hierarchical Encoding-based Neural Representation"
_NeurIPS.cc/2023/Conference — NeurIPS 2023 poster_

### Official Review · Reviewer_H9aP · 2023-07-05

**Soundness:** 3 good
**Presentation:** 3 good
**Contribution:** 3 good
**Rating:** 5
**Confidence:** 4

**Summary:**

This paper presents HiNeRV, a new INR model for video compression based on Hierarchically-encoded Neural Representation. HiNeRV introduces a new upsampling layer that replaces conventional sub-pixel layers in existing INRs. This layer combines bilinear interpolation with hierarchical encoding, utilizing multi-resolution local feature grids. Additionally, HiNeRV incorporates depth-wise convolutional and MLP layers to construct a deep and wide network architecture with higher capacity. Sufficient experiments on several datasets and tasks have demonstrated the effectiveness of the proposed method.

**Strengths:**

-	The ideas in this paper appear to be novel (to my knowledge) and straightforward and should be easily re-implement given the paper’s descriptions.
-	The authors have made many technical improvements and engineering efforts to achieve competitive coding performance.
-	Extensive experiments have been done to demonstrate the effectiveness and compactness of the proposed method. Although HiNeRV has not been end-to-end optimized for video compression, it still significantly outperforms previous approaches with comparable bit rates.


**Weaknesses:**

-       There are several figures that may be improved for better illustration to the readers. In particular, I don't like Figure 2 since it seems too involved and without any summary in the caption.
-	Although the proposed method outperforms other methods in terms of encoding quality, the proposed method introduces more computational complexity, which makes encoding and decoding slower and is not conducive to real-time applications.
-	There are some errors in the references, for example, reference 9 should be “HNeRV: a hybrid neural representation of videos” instead of  “HNeRV: neural representations for videos”. Please check your citations.


**Questions:**

-	I am confused why the performance of HNeRV will be worse than NeRV performance, and whether the experiment is correct.
-	Limitations or bad case discussion and analysis are expected.
-	Other concerns have already been mentioned in Weakness.


**Limitations:**

	Yes, the authors have addressed the limitations and potential negative societal impact of their work.

---

> ### Author Rebuttal · Authors · 2023-08-09
>
> **Q: The figures require better illustration.**
>
> A: We will re-draw these figures and provide a more detailed description in the caption for improved clarity.
>
>
> **Q: The proposed method does not fit real-time application.**
>
> A: We agree that encoding and decoding are not yet sufficiently fast for real-time applications. However, it should be noted that HiNeRV achieves a much faster decoding speed than other learning-based codecs (such as DCVC/DCVC-HEM) as mentioned in the supplementary material (Section E). Furthermore, see Section D.3 of the supplementary material, we show that, by adjusting the architecture, HiNeRV has a similar decoding speed to HNeRV (up to 56.9 FPS with a modern GPU) with both better compression performance and lower theoretical complexity.
>
> The runtime performance of the proposed model can be further improved by optimizing the implementation, inference with lower precision, structural pruning, etc. We will highlight this in the revised paper as our future work.
>
>
> **Q: Errors in the references.**
>
> A: Thank you for pointing out these issues. We will check all the citations thoroughly.
>
>
> **Q: Performance issue of HNeRV.**
>
> A: There are multiple possible reasons leading to this issue. In our experiments, we use both of the official implementations and configurations of NeRV and HNeRV. We notice that the configuration of NeRV in the NeRV repository is different to that of NeRV in the HNeRV repository. Specifically, the configuration of NeRV in the NeRV repository limits the minimum width of the convolutional layers to 96, where such width is only 12 in both configurations for NeRV/HNeRV in the HNeRV repository. Setting the minimum width to a larger value can lead to a better performance, as more computation will be done in higher resolutions. This could contribute to part of the performance difference between our results and the HNeRV paper.
>
> Also, we have different experiment settings. Due to the stride ratios, HNeRV with the original configuration cannot represent videos with 1920x1080 resolution. In the original HNeRV paper, the authors crop the videos to 1920x960, while we perform padding to make HNeRV compatible with 1920x1080 resolution.
>
> Furthermore, in the HNeRV paper, the authors did not provide results of HNeRV for bpp $>$ 0.06 in the UVG dataset; we observed that NeRV performs better than HNeRV in this extended range.
>
> Lastly, we observed that HNeRV is not stable when training with float16 for some sequences in the MCL-JCV dataset, as mentioned in Section A of the supplementary material. When training with float32, the performance of HNeRV can be improved, but still incomparable with HiNeRV. We will update the result of HNeRV with float32 for a fair comparison. We also reported the result shown in the global response section (Figure 1) with the updated numbers.
>
>
> **Limitation and worst case**
>
> We will include a discussion regarding the limitations and the worst cases in  the revised paper.
>
> Limitations:
> Although our work has shown promising results in compression performance, similar to existing INR base approaches, the encoding time is still a limitation. The encoding time is due to the requirement for model training. Reducing encoding time will be an important direction for future work.
>
> Worst cases:
> In the global response, we have provided the best/worst case (PSNR) performance for HiNeRV in the UVG/MCL-JCV datasets.  For the UVG dataset, we found that HiNeRV performs worst for the Beauty sequence; however the performance is still far better than for other INR based methods including NeRV and HNeRV, as well as for x265 (veryslow). For the MCL-JCV dataset, HiNeRV perform worst with the videoSRC08 sequence. In this sequence, HiNeRV is not as good as x265 (veryslow), but still far better than NeRV/HNeRV.
>
> For both cases we found that HiNeRV still obtains superior performance in terms of MS-SSIM outperforming most baselines including HM (randomaccess) in the Beauty sequence, and comparable to HM (randomaccess) in the videoSRC08 sequence.
>
> Also, as pointed out by reviewer JKXU, HNeRV has poor performance in representing high dynamic scenes. In Section 4.1 Table 2 of the paper, we show that HiNeRV has an average of 8.35 and 6.95 dB PSNR improvement over NeRV and HNeRV, respectively, for the highly dynamic scene for which HNeRV performed worst (ReadySetGo). Also, in contrast to HNeRV, we did not observe any clear disadvantages in representing specific sequence classes in HiNeRV.

---

### Official Review · Reviewer_zJ4f · 2023-07-05

**Soundness:** 3 good
**Presentation:** 3 good
**Contribution:** 2 fair
**Rating:** 5
**Confidence:** 4

**Summary:**

This paper presents a novel method for video compression using implicit neural representation (INR), where video information is stored in the model weights. The authors explore various advanced techniques to enhance the architecture's capabilities. The primary contribution is the utilization of bilinear interpolation with hierarchical encoding for upsampling, which enables the construction of a higher-capacity network compared to using sub-pixel convolutional layers. Experimental results demonstrate that the proposed method outperforms existing INR models and is the first INR-based video codec to outperform HEVC.

**Strengths:**

- This paper is well-organized and easy to follow.

- The authors achieve state-of-the-art INR-based video compression performance, particularly surpassing HEVC for the first time.

- Detailed experimental settings and comparisons are provided.

**Weaknesses:**

- The authors do not provide the source code.

- The comparison with FFNeRV, which also exhibits superior performance to HEVC, is missing.

- The analysis of different interpolation methods, such as nearest neighbor and linear interpolation, is not included

**Questions:**

- Is the architecture the same for videos with a different number of frames? If not, how is the model's size adjusted?

- Please include a comparison with FFNeRV, as it also outperforms HEVC.

**Limitations:**

Yes

---

> ### Author Rebuttal · Authors · 2023-08-09
>
> **Q: Source code is not provided.**
>
> A: As mentioned in the supplementary material (line 4), we will provide the full implementation of HiNeRV when the paper is accepted.
>
>
> **Q: Comparison to FFNeRV is missing.**
>
> A: We did provide benchmark results of FFNeRV as shown in Section 4.1 (Table 1-2) for the video representation task.
>
> We will also include its results for video compression in the revised version, as shown in the global response section (Figure 1). It is noted that FFNeRV is only comparable with the HEVC x265 with medium preset, where HiNeRV outperforms the HEVC x265 with the veryslow preset and is comparable with the HEVC HM with the randomaccess profile. In this case, we are confident in our claim that HiNeRV performs better than FFNeRV for the video compression task as well.
>
>
> **Q: Analysis of different interpolation methods is missing.**
>
> A: Following the reviewer's suggestion, we will provide the comparison results with the nearest neighbor up-sampling filter in the revised paper, as shown in the global response section (Table 1). Our results have shown that, even with nearest neighbor up-sampling, HiNeRV still exhibits superior performance, an average PSNR that is only slightly worse than with the default settings.
> It is noted that, as we are using the same spatial grid size for both the first level feature maps and the grids, the employed trilinear interpolation is effectively the same as performing linear interpolation in the temporal dimension.
>
>
> **Q: How to adapt the network to different video lengths?**
>
> A: In our experiments, we share the same configurations for all videos in the same dataset for simplicity. For example, we use the same configurations for all 7 videos in the UVG dataset, although there are videos with 300/600 frames. For different datasets (e.g., UVG vs MCL-JCV), we did change the hyper parameters of the model, as provided in the supplementary material (Table 1)

---

> > ### Comment · Reviewer_zJ4f · 2023-08-18
> >
> > Thank you for the authors' efforts in the rebuttal. Their response has addressed my concerns, and I have increased my rating.

---

### Official Review · Reviewer_JKXU · 2023-07-07

**Soundness:** 2 fair
**Presentation:** 2 fair
**Contribution:** 2 fair
**Rating:** 5
**Confidence:** 4

**Summary:**

This paper presents a new type of implicit neural representations (INR) with hierarchical positional encoding for video compression, named HiNeRV. Four points are proposed: (1) a new upsampling layer with bilinear interpolation and hierarchical encoding of feature grids, (2) the use of depth-wise convolution and MLP layers instead of conventional convolution layers, (3) the unified representation of frame-wise and patch-wise INR by adding padding, and (4) refined training pipeline for HiNeRV model compression. The proposed method achieves significant improvement over all existing INRs baselines and competitive performance to recent neural video codecs.

**Strengths:**

Significant rate-distortion performance as an INR-based video codec. Largely closing the gap between INR-based method and the SOTA neural video codecs.

**Weaknesses:**

Although the proposed method achieves very significant video compression performance, the motivations behind the major contributions are not clearly discussed. Specifically, as the origin of HiNeRF, the hierarchically-encoded neural representation comes from the proposed new upsampling layer with hierarchical encoding on local feature grids. So why this design provides better performance? The paper claims that this design is similar to a convolutional layer over constant feature map. As one of the major contributions, it would be better to provide more comprehensive discussion and comparison.

Moreover, according to the ablation studies, the major performance gain compared to previous methods seems to mainly come from the use of bilinear interpolation, which is a little bit incremental.

Finally, it would be better to improve the paper writing. For example, as one of the contributions, the unified frame-wise and patch-wise representation is not mentioned in the abstract. And the analysis in ablation studies can be further improved.


**Questions:**

Below are some further questions:
(1)	Why the proposed method performs a bit worse on MCL-JCV dataset?
(2)	Is it possible to provide the best/worst compression cases? In HNeRV, the authors show their model has limited performance on highly dynamic scenes. Is there a similar issue in HiNeRV?
(3)	Is it possible to provide the comparison of encoding complexity with recent neural video codecs?
(4)	How do you compress the feature grid? How much bitrate cost of the feature grid?
(5)	Some typos in the paper, e.g. Line 60: before quantiszation -> before quantization

---

> ### Author Rebuttal · Authors · 2023-08-09
>
> **Q: The motivations of the major components are not clear.**
>
> A: The advantage of the hierarchical encoding can be considered from different perspectives:
> Firstly, existing works have shown that positional encoding is helpful in learning high frequency information. In HiNeRV, when performing upsampling with the bilinear interpolation layer, it generates high dimensional but smoothed feature maps. By introducing the hierarchical encoding, we are able to encoded the relative positional information, which complements the interpolation layer. In our ablation studies, we found HiNeRV with MLP block still obtain acceptable performance, which is different from observations reported in previous works (the NeRV paper). We expect that this is primarily due to the use of the hierarchical encoding.
>
> When combining the input feature grids with multiple local feature grids (for extracting the hierarchical encoding) at different scales, we are able to represent different spatial locations in a hierarchical manner, which is a more efficient representation than using high dimension 'global' grids. For example, instead of having a global grid of size $T \times H \times W \times C$, we can have a global grid with multiple local grids with a total size of $\frac{T}{R} \times \frac{H}{S_0 \times S_1 \times ...} \times \frac{W}{S_0 \times S_1 \times ...} \times C + T \times S_0 \times S_0 \times C + ...$, where $R$ is a reduction factor to reduce the temporal resolution of the `global' grid, and the $S_0, S_1, ...$ are the up-scaling ratios, which are also the spatial sizes of the local grid as we use the local grid to encode the relative positional information during up-scaling. It also allows us to represent temporal signals at a higher resolution, given the same parameter budget. Our ablation studies have shown that the hierarchical encoding specifically benefit video sequences with fast motion (Jockey/ReadySetGo).
>
> To make these points clearer, we will provide detailed justification regarding the motivation of our designs in the revised paper.
>
>
> **Q: The major gain is coming from the use of bilinear interpolation.**
>
> A: While we agree that bilinear interpolation plays an important role in our work, it can also be seen that other elements in HiNeRV (encoding, choice of block and unified representation) also affect the performance significantly. Specifically, we found that encoding helps with videos with fast motion -  e.g. improving the PSNR of Jockey/ReadySetGo by 0.78/1.06 dB. With the default choice (the ConvNeXT block) HiNeRV produces an increase of 1.15 dB in terms of the average PSNR compared to using normal convolutional blocks. By training as a unified representation, while still performing computation in patches, HiNeRV can obtain a 0.89 dB PSNR gain over a normal patch-wise representation.
>
> We will make this point clearer in the revised paper.
>
>
> **Q: The paper writing has to be improved.**
>
> A: We commit to fully revising the writing of the paper to improve clarity and to include the use of the unified representation as one of the contributions in the abstract. We will also provide a more detailed discussion regarding the results of the ablation studies.
>
>
> **Q: Why the performance is worse on MCL-JCV dataset?**
>
> A: This is an interesting point and may be due to the different video frame-rates. In the UVG dataset, videos are captured at 120fps, whereas the frame rates for MCL-JCV are 25-30fps. INR-based approaches may perform better for content with more temporal redundancy compared to the anchors. We will comment on this in the revised paper.
>
>
> **Q: What is the best/worst compression cases?**
>
> A: Again, a good point. We will provide examples of best/worst cases in the revised paper, as shown in Fig 2-3 of the global response section. For HiNeRV, the best/worst performances are for the HoneyBee/Beauty sequences in the UVG dataset, and the videoSRC29/videoSRC08 sequences in the MCL-JCV dataset, respectively. We also found that, despite having worse PSNR performance than conventional codecs in the worst cases, it still outperforms all previous INR-based methods including NeRV and HNeRV in most cases. We also provide MS-SSIM results for these sequences in the global response, where HiNeRV outperforms both of the conventional codecs (HM (Random access) and x265 (veryslow)) and INR-based methods for most rates.
> Unlike HNeRV, we found that HiNeRV also performs well in representing highly dynamic scenes. As we can see from the results in the submitted paper (Table 2), HiNeRV outperforms HNeRV significantly on the ReadySetGo sequence.
>
>
> **Q: Compare encoding complexity with neural codes**
>
> A: We will provide more details regarding the encoding complexity in the revised paper, as shown in Table 2 in the general response section. And we will also provide comparison with neural video codecs in the revised paper.
>
> We emphasize that encoding complexity is currently one of the major limitations of INR-base methods, due to the need for model training. However, INRs typically achieve better decoding speed. As mentioned in the supplementary materials (Section E), HiNeRV can achieve up to 10.9 FPS with the scale L (which corresponds to the middle/largest model for UVG/MCL-JCV, respectively), where other state-of-the-art neural codecs have a much slower decoding speed, e.g. DCVC-HEM can only achieve 1.9 FPS.
>
>
> **Q: How to compress the feature grid? How large is it?**
>
> A: We consider feature grids as normal parameters in this work, simply calculating the frequency of the quantized symbols and performing arithmetic coding. They contribute less than 20\% of the total bitrate. We will revise the paper and provide this information.
>
>
> **Q: Typos in the paper.**
>
> A: Thank you for pointing this out. We will fix these typos and check throughout the paper.

---

> > ### Comment · Reviewer_JKXU · 2023-08-18
> >
> > Thanks for the author’s detailed response and more new results. Some of my concerns have been addressed. Although it still requires a major change from the initial version, I am going to increase my rating.

---

### Official Review · Reviewer_zt3G · 2023-07-07

**Soundness:** 3 good
**Presentation:** 2 fair
**Contribution:** 3 good
**Rating:** 5
**Confidence:** 5

**Summary:**

The authors propose a 2D implicit neural compression method that combines bilinear interpolation and hierarchical encoding. This method is based on the premise that bilinear interpolation is a better choice for neural compression than convolution operations.

**Strengths:**

1. The proposed method distinguishes itself from other existing methods by utilizing bilinear interpolation for feature upsampling.

2. The paper investigates model compression techniques, including pruning and quantization, to further reduce the size of the model.

3. The effectiveness of the proposed modules is convincingly demonstrated through an ablation study.

**Weaknesses:**

1. The explanation of the hierarchical coding process in section 3.4 and Fig.2 is unclear and difficult to follow, requiring further clarification.
2. It is crucial for the paper to include results on encoding and decoding time, as this information holds significant relevance in real-world scenarios.
3. The use of deep models with overlapped patches may lead to an unfair comparison with other implicit compression methods due to the increase in training pixels it implies. However, the paper lacks sufficient explanation on this matter, particularly regarding how the final reconstructed pixels are determined by overlapping blocks.

**Questions:**

Does the training method in this paper involve training only one model on the entire dataset for inference, as mentioned in NeRV[10]? How does this encoding/decoding process apply to actual scenarios? If so, how is the compression performance on a single-sequence dataset like bunny?

**Limitations:**

While the paper effectively improves the compression efficiency of the implicit codec, there is scope for future research in optimizing encoding time and latency.

---

> ### Author Rebuttal · Authors · 2023-08-09
>
> **Q: The explanation of the hierarchical coding is not clear**
>
> A: In the revised paper, we will modify the relevant text to better describe the hierarchical coding process. We will also re-draw Fig. 2 to make it easier to understand. Furthermore, we will release the code when paper is accepted, which will help others to understand the implementation details.
>
>
> **Q: Encoding/decoding time is not given**
>
> A: We agree with this reviewer regarding the importance of reporting the encoding/decoding time in the paper. Actually we did report encoding time in the supplemental material for one scale (line 140). We will include more comprehensive encoding runtime results in the revised version, as provided in the global response section (Table 2), where the detailed encoding time comparison results between different INRs are summarised.  For the decoding time results, we have already provided these in the main paper (Table 1-2, in FPS).
>
>
> **Q: The overlapped patches lead to unfair comparison**
>
> A: Although overlapped patches provide performance improvements for HiNeRV, HiNeRV still significantly outperforms NeRV and HNeRV without them. In the global response section (Table 1), we provide more additional ablation results, showing that HiNeRV outperforms NeRV and HNeRV with either frame or patch configurations.
>
>
> **Q: How to obtain final pixels from the overlapped patches**
>
> A: Regarding how to determine the final reconstructed pixels, we did briefly mention this in the supplementary materials (Line 43). The final reconstructed pixels are simply obtained by center cropping, as the amount of padding is equal on each side. To clarify this, we will include a more detailed description in the revised paper.
>
>
> **Q: Do the networks encode the entire dataset/individual videos?**
>
> A: In our work, we train one model for each video (as mentioned in the main paper, Line 305), which differs from the NeRV paper. This, in fact, makes HiNeRV more practical. In Section 4.1 and Table 1 of the main paper, we provide HiNeRV results for the Bunny dataset. HiNeRV is able to outperform all baselines significantly. For example, HiNeRV with 0.77M parameters is able to obtain a reconstruction quality to very close to HNeRV with 3.28M parameters.
>
>
> **Q: Future research of encoding time and latency**
>
> A: We totally agree that reducing the encoding time and latency will be an important direction for future work. We will mention this in the revised paper.

---

> > ### Comment · Reviewer_zt3G · 2023-08-19
> > **Improved Performance**
> >
> > Thank you for the author's response, it clarified some issues. In my opinion, the main contribution of this paper is indeed a relatively small but also clever improvement in terms of technique, which brings noticeable performance gains to the overall system. One major issue with this paper is its writing aspect, I believe it needs substantial revisions. The proposed hierarchical encoding does not appear to have much novelty from my perspective, although this term is frequently mentioned. Overall, there are strong advantages in terms of performance but lacking novelty and requiring further improvements in writing. I will maintain my current score.

---

> > > ### Author Response · Authors · 2023-08-21
> > >
> > > We appreciate the additional comments from Reviewer zt3G. Please note that these issues have also been raised by other reviewers, and we have addressed them in the previous responses. Regarding novelty, please refer to our response to reviewer boCg, and for issues related to the novelty of (and motivation for) hierarchical encoding, we have clarified addressed this in our response to reviewer JKXU.
> > >
> > > For convenience, we summarize our replies and provide some additional information here.
> > >
> > > **The overall novelty**
> > >
> > > 1\. We utilized parameter efficient layers for INR-based video compression. In INR-based compression, the number of parameters is directly related to the compression performance. However, all previously published work has employed layers with high complexity, e.g. sub-pixel convolution. In contrast, in HiNeRV, we use a bilinear interpolation layer, depth-wise convolutional layers and MLPs as the network layers. Despite the simplicity of this innovation, we have clearly shown that significant gains can be obtained by adopting parameter efficient layers.
> > >
> > > 2\. We also proposed hierarchical encoding, which encodes the relative positional information. Hierarchical encoding is parameter efficient as it scales with the upsampling ratio of the layer, rather than the full spatial dimensions. By adding multiple hierarchical encodings at different layers, we can represent high resolution signals, without the need for feature grids with high storage cost. When fixing the number of parameters, hierarchical encoding provided significant improvement for highly dynamic video sequences where previous works (e.g. NeRV/HNeRV) performed poorly (0.78/1.06 dB in PSNR for the Jockey/ReadySetGo sequence).
> > >
> > > In our reply to reviewer JKXU, we mentioned that the high performance of HiNeRV with the MLP block is due to the use of the hierarchical encoding. To validate this, we performed an additional experiment and found that if we remove the hierarchical encoding from HiNeRV (with MLP blocks) then the average PSNR decreased by 2.54 dB. HiNeRV with MLP blocks, each of which only has a very lightweight convolutional layer in the stem, clearly outperformed the state-of-the-art CNN-based approach, HNeRV. The high performance of the MLP-based network is also different from the observation in previous work (the NeRV paper), which reported that MLP-based networks perform poorly in the context of the video encoding task. This further confirms the effectiveness of hierarchical encoding. We believe the use of the hierarchical encoding could also benefit MLP-based INR, but this remains for future research.
> > >
> > > 3\. While existing INR approaches for video compression are either frame- or patch-based, we introduced a unified representation. Our proposed HiNeRV, as a unified representation, can thus be utilized with either frames or overlapped patches. It yields better performance than a frame-wise representation, but also benefits from the parallelism of a patch-wise representation which can reduce the minimum memory requirement for computation. The performance improvement is confirmed in the ablation study (Section 4.3 in the paper), where we observe that  HiNeRV with the unified representation obtained up to 0.49 dB PSNR improvement over the frame-wise configuration, on the UVG dataset.
> > >
> > > We also found that the improvement is more significant when the encoding time is reduced. We performed an additional experiment following the settings used in Section 4.1 with only 37 epochs, and we observed that HiNeRV with the unified representation obtained 0.93 dB improvement in average PSNR compared with the frame-wise configuration. Moreover, with our unified representation, we can perform computation using patches without sacrificing performance -  an important characteristic for INR-based video compression, due to the higher signal dimensionality. For example, a recently published paper [1] had to divide the videos into patches due to memory constraints.
> > >
> > > **Writing of the paper**
> > >
> > > We appreciate the comments provided by the reviewers regarding the paper's writing. We will revise the paper in order to enhance its clarity and readability.
> > >
> > > [1] He et.al, Towards Scalable Neural Representation for Diverse Videos, CVPR 2023

---

### Official Review · Reviewer_boCg · 2023-07-08

**Soundness:** 3 good
**Presentation:** 2 fair
**Contribution:** 2 fair
**Rating:** 5
**Confidence:** 4

**Summary:**

This paper introduces a hierarchical encoding block for INR-based video compression. In particular, the authors replace the commonly-used sub-pixel/transpose convolution for upsampling with the bi-linear interpolation. The saved parameters are used to increase the capacity of the MLP encoders in order to achieve higher expressiveness of the model. Reportedly, this work is the first INR-based method that achieves comparable rate-distortion performance to the state-of-the-art VAE-based learned methods.

**Strengths:**

(1) The paper is easy to follow.
(2) The coding results are promising.

**Weaknesses:**

(1) The novelty of this paper is rather now. Essentially, the authors replace the commonly-used sub-pixel/transpose convolution for upsampling with the bilinear interpolation. In doing so, the saved parameters are used to increase the number of MLP encoders. The gain appears to come mainly from a simple tweak to the network architecture.

(2) The network architecture of the proposed method is not new, and is more like a trivial combination of several existing works. For example, the Grid and Stem modules are from [1] and [2]. After index transformation, the feature upsampling part is from [3].

[1] J. C. Lee, D. Rho, J. H. Ko, and E. Park. FFNeRV: Flow-guided frame-wise neural representations for videos. arXiv preprint arXiv:2212.12294, 2022.

[2] J. L. Ba, J. R. Kiros, and G. E. Hinton. Layer normalization. arXiv preprint arXiv:1607.06450, 2016

[3] Z. Liu, H. Mao, C.-Y. Wu, C. Feichtenhofer, T. Darrell, and S. Xie. A convnet for the 2020s. Proceedings of the IEEE/CVF Conference on Computer Vision and Pattern Recognition (CVPR), 2022.

(3) The authors should include the comparison with DCVC-HEM in the main paper. Not sure why DCVC (which is a weaker baseline) is used instead. Also, please clarify the GOP size. It seems that GOP=12 is adopted for DCVC-HEM. It was argued in several papers (including DCVC-HEM) that a larger GOP should be used.

(4) The comparison with HNeRV may be unfair, because the network of HNeRV is designed to represent the whole dataset. As such, its network is quite heavy, and can actually be reduced considerably when it is used to represent only one sequence. In this sense, the per-sequence rate reported for HNeRV may be overestimated.

[4] H. Chen, M. Gwilliam, S.-N. Lim, and A. Shrivastava. HNeRV: Neural representations for videos. In CVPR, 2023.

**Questions:**

See my comments in the weaknesses section.

**Limitations:**

The encoding time is not addressed.

---

> ### Author Rebuttal · Authors · 2023-08-09
>
> **Q: Novelty of the paper**
>
> A: The novelty of this work is three-fold. (i) Starting with NeRV, all INR-based codecs employ sub-pixel convolution; this is one of the main limitations that has not been addressed yet. In this work, we introduce parameter efficient layers: this is a simple but effective idea which is shown to be critical for INR based video compression. (ii) While this modification of the architecture does contribute to the overall coding gains achieved, we emphasise that our use of hierarchical encoding also offers significant gains on some video sequences (Jockey/ReadySetGo), where existing INRs (including NeRV and HNeRV) perform poorly. (iii) Finally, we proposed a unified representation, which provides additional parallelism alongside performance gains.
>
> We will revise the text in the paper to make the above contributions clearer. Furthermore, in order to better characterise the significance of each contribution, we will provide more comprehensive ablation study results including those from NeRV and HNeRV for comparison, as shown in the global response section (Table 1). It can be observed that, when replacing the bilinear interpolation in HiNeRV with a sub-pixel convolutional layer with 1x1/3x3 kernel size (Variant 1/2), HiNeRV still outperforms NeRV. When comparing with HNeRV, the variant with 1x1 kernel size is superior, while the variant with 3x3 kernel size is comparable with it based on PSNR (the MS-SSIM results are even better, i.e., 0.9664 vs 0.9382). We also note that HNeRV employs an additional encoder network and it is tweaked heavily -  the authors tuned the reduction ratio and kernel sizes across layers to obtain better performance.
>
> It should also be noted in Table 1 of the global response and from the ablation study results in the original paper that other elements in HiNeRV (encoding, choice of block and unified representation) also affect the performance significantly. Specifically, we found that encoding helps more with videos that exhibit fast motion - e.g. it improves the PSNR of Jockey/ReadySetGo by 0.78/1.06 dB. With the default choice (using the ConvNeXT block) HiNeRV obtain delivers an increase of 1.15 dB in terms of average PSNR compared to using normal convolutional blocks. By training as a unified representation, while still performing computation in patches, HiNeRV obtains a 0.89 dB PSNR gain over a normal patch-wise representation.
>
> All these results clearly demonstrate that all the design elements contribute to the overall performance improvements in HiNeRV.
>
>
> **Q: Novelty of the architecture**
>
> A: We agree that the Grid/Stem modules and feature upsampling are not new -- we do not claim these as novel contributions in this work. Instead, as  mentioned above, the novelty of this paper is based on: (i) applying a parameter efficient network architecture for video compression, (ii) hierarchical encoding and (iii) a unified representation. All of these contribute to the overall performance of HiNeRV based on the ablation study results provided in the global response section (and in the paper). Detailed justification can be found above.
>
>
> **Q: Comparison with DCVC-HEM**
>
> A: Following the suggestion of the reviewer, we will include the comparison with DCVC-HEM in the main paper (also shown in the global results section of the rebuttal). For DCVC-HEM, we use GOP=32. Please note that, in our paper, we are using full video sequences for evaluation rather than 96 frames as in the original DCVC-HEM paper. This may lead to different results.
>
>
> **Q: Comparison with HNeRV**
>
> A: The authors of HNeRV have specified that their experiments were performed by training one model for one video. Quoting from the HNeRV paper: \textit{``Note that HNeRV achieves this using a small model for each video, while NeRV fits a big model on concat videos (for better compression) which greatly slows down the encoding and decoding speed."}. In our experiment, we always use one model for one video for both HiNeRV and baseline models in order that their rate won't be overestimated.
>
>
> **Q: The encoding time is not addressed**
>
> A: We did report the encoding time in the supplementary material associated with the submitted paper. Specifically, HiNeRV with scale L takes around 12 hours for encoding 1080p videos with 600 frames. This is the time required for the full training following the experimental settings for video compression. We will extend this to provide the estimated encoding time comparison of different INRs with the UVG dataset in the revised paper, as shown in the global response section of the rebuttal (Table 2). The results are based on 300 epochs for training with 90 epochs of fine-tuning. Although HiNeRV is slower in terms of encoding time, the faster variant, HiNeRV-B, which we also reported in the supplementary material, is able to perform faster encoding with a smaller model. For example, HiNeRV-B (scale-M) is faster than HNeRV (scale-L), while the model size is smaller and the reconstruction quality is better (reported in Table 5 of the supplementary material).
>
> HiNeRV is also able to obtain superior reconstruction quality in a much shorted time, as shown in Figure 1 (right) of the main paper.

---

> > ### Comment · Reviewer_boCg · 2023-08-16
> > **Comparison with DCVC-HEM**
> >
> > I thank the authors for putting effort into addressing my comments. Regarding the comparison with DCVC-HEM, it seems that the proposed method still encodes the entire sequence instead of 96 frames. This makes the comparison inconclusive.

---

> > > ### Author Response · Authors · 2023-08-16
> > >
> > > We thank the reviewer for the further comment. To clarify, we conducted the experiments using full length video sequences. This ensured that the results were fairly compared and more representative. Hence both HiNeRV and DCVC-HEM were evaluated under identical conditions, using complete video sequences. We therefore consider our comparison to be fair and conclusive.

---

> > > > ### Comment · Reviewer_boCg · 2023-08-22
> > > >
> > > > Thanks for the clarification. I would change my rating to weak accept.

---

### Author Rebuttal · Authors · 2023-08-09

We thank all the reviewers for reviewing and providing detailed feedback to our submission. We will address the concerns individually.

---

### Decision · Program_Chairs · 2023-09-21

**Decision:**

Accept (poster)

**Comment:**

This paper presents a simple method to improve INR-based video compression by introducing lightweight architectural changes and heirarchical positional encoding. It further investigates pruning and quantization to reduce the size of the model. These changes are validated by extensive experiments and comparisons. The authors addressed most reviewer concerns, and reviewers agreed that this work should be accepted by the end of the discussion phase.